# MODULI: Unlocking Preference Generalization via Diffusion Models for Offline Multi-Objective Reinforcement Learning

Yifu Yuan [* 1]   Zhenrui Zheng [* 1 2]   Zibin Dong [1]   Jianye Hao [1]

## Abstract

Multi-objective Reinforcement Learning (MORL) seeks to develop policies that simultaneously optimize multiple conflicting objectives, but it requires extensive online interactions. Offline MORL provides a promising solution by training on pre-collected datasets to generalize to any preference upon deployment. However, real-world offline datasets are often conservatively and narrowly distributed, failing to comprehensively cover preferences, leading to the emergence of out-of-distribution (OOD) preference areas. Existing offline MORL algorithms exhibit poor generalization to OOD preferences, resulting in policies that do not align with preferences. Leveraging the excellent expressive and generalization capabilities of diffusion models, we propose **MODULI** (**M**ulti **O**bjective **D**iff**U**sion planner with s**LI**ding guidance), which employs a preference-conditioned diffusion model as a planner to generate trajectories that align with various preferences and derive action for decision-making. To achieve accurate generation, MODULI introduces two return normalization methods under diverse preferences for refining guidance. To further enhance generalization to OOD preferences, MODULI proposes a novel sliding guidance mechanism, which involves training an additional slider adapter to capture the direction of preference changes. Incorporating the slider, it transitions from in-distribution (ID) preferences to generating OOD preferences, patching, and extending the incomplete Pareto front. Extensive experiments on the D4MORL benchmark demonstrate that our algorithm outperforms the state-of-the-art Offline MORL baselines, exhibiting excellent generaliza-

tion to OOD preferences. Codes are available on this repository.

## 1. Introduction

Real-world decision-making tasks, such as robotic control (Liu et al., 2014; Cui et al., 2025; Yuan et al., 2025), autonomous driving (Li & Czarnecki, 2018), and industrial control (Bae et al., 2023), often require the optimization of multiple competing objectives simultaneously. It necessitates the trade-offs among different objectives to meet diverse preferences (Roijers et al., 2017; Yuan et al., 2024; Liu et al., 2024; Zhou et al., 2025). For instance, in robotic locomotion tasks, users typically focus on the robot's movement speed and energy consumption (Xu et al., 2020). If the user has a high preference for speed, the agent will move quickly regardless of energy consumption; if the user aims to save energy, the agent should adjust to a lower speed. One effective approach is Multi-objective Reinforcement Learning (MORL), which enables agents to interact with vector-rewarded environments and learn policies that satisfy multiple preferences (Hayes et al., 2022; Roijers et al., 2013; Yang et al., 2019). These methods either construct a set of policies to approximate the Pareto front of optimal solutions (Alegre et al., 2023; Felten et al., 2024) or learn a preference-conditioned policy to adapt to any situation (Lu et al., 2023b; Hung et al., 2022). However, exploring and optimizing policies for each preference while alleviating conflicts among objectives, requires extensive online interactions (Alegre, 2023), thereby posing practical challenges due to high costs and potential safety risks.

Offline MORL (Zhu et al., 2023a) proposes a promising solution that can learn preference-conditioned policies from pre-collected datasets with various preferences, improving data efficiency and minimizing interactions when deploying in high-risk environments. Most Offline MORL approaches focus on extending return-conditioned methods (Thomas et al., 2021; Zhu et al., 2023a) or encouraging consistent preferences through policy regularization (Lin et al., 2024). However, real-world offline datasets often come from users with different behavior policies, making it difficult to cover the full range of preferences and leading to missing prefer-

---

[*]Equal contribution [1]College of Intelligence and Computing, Tianjin University [2]Harbin Engineering University. This work was done when ZZ was visiting Tianjin University. Correspondence to: Jianye Hao <jianye.hao@tju.edu.cn>.

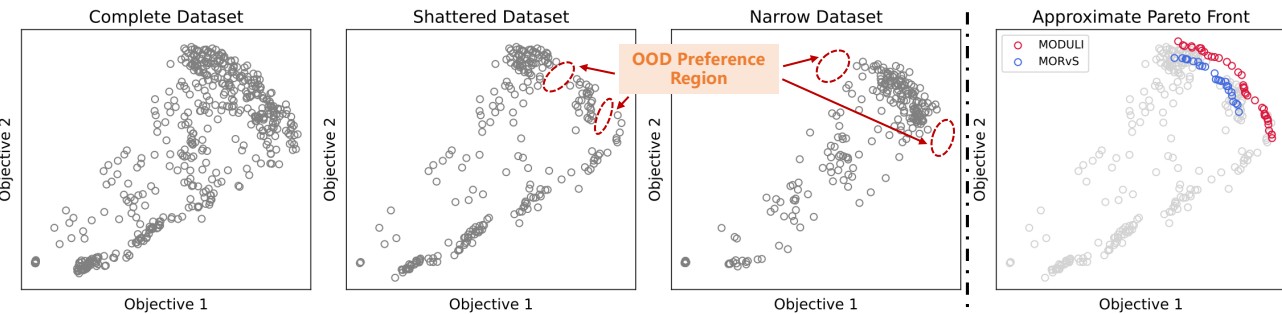

*Figure 1.* (**Left**) Trajectory returns for `Hopper-Amateur` datasets in D4MORL. We visualize `Complete`, `Shattered`, and `Narrow` datasets for comparison. OOD preference regions are marked by the red circles. (**Right**) The approximated Pareto fronts learned by MORvS and MODULI on the `Shattered` dataset.

ence regions, i.e., OOD preferences, resulting in a shattered or narrow data distribution. As shown in Figure 1, we visualize the trajectory returns for `Hopper-Amateur` dataset. The `Complete` version of the dataset can comprehensively cover a wide range of preferences. In many cases, suboptimal policies can lead to `Shattered` or `Narrow` datasets, resulting in OOD preference regions as marked by the red circles. We then visualize the approximate Pareto front for MORvS baseline (Zhu et al., 2023a) and our MODULI in such a shattered dataset. We find that the current Offline MORL baselines exhibit poor generalization performance when learning policies from incomplete datasets. They fail to adapt to OOD preference policies. The approximate Pareto front of MORvS shows significant gaps in the OOD preference areas, indicating that the derived policies and preferences are misaligned. Therefore, we aim to derive a general multi-objective policy that can model a wide range of preference conditions with sufficient expressiveness and strong generalization to OOD preferences. Motivated by the remarkable capabilities of diffusion models in expressiveness and generalization, we propose **MODULI** (**M**ulti **O**bjective **D**iff**U**sion planner with s**LI**ding guidance), transforming the problem of Offline MORL into conditional generative planning. In Figure 1 (right), MODULI demonstrated a better approximation of the Pareto front, achieving returns beyond the dataset. MODULI also patched the missing preference regions and exhibited excellent generalization for the OOD preferences.

MODULI constructs a multi-objective conditional diffusion planning framework that generates long-term trajectories aligned with preference and then executes as a planner. Furthermore, appropriate guiding conditions play a crucial role in the precise generation, which requires the assessment of trajectory quality under different preferences for normalization. We found that simply extending the single-objective normalization method to multi-objective causes issues, as it fails to accurately align preferences with achievable high returns. Therefore, we propose two return normalization

methods adapted to multi-objective tasks, which can better measure the quality of trajectories under different preferences and achieve a more refined guidance process. Besides, to enhance generalization to OOD preferences, we introduce a sliding guidance mechanism that involves training an additional slider adapter with the same network structure to independently learn the latent directions of preference changes. During generation, we adjust the slider to actively modify the target distribution, gradually transitioning from ID to specified OOD preferences. This better captures the fine-grained structure of the data distribution and allows for finer control, thereby achieving better OOD generalization. In summary, our contributions include (**i**) a conditional diffusion planning framework named MODULI, which models general multi-objective policies as conditional generative models, (**ii**) two preference-return normalization methods suitable for multi-objective preferences to refine guidance, (**iii**) a novel sliding guidance mechanism in the form of an adapter and, (**iv**) superior performance in D4MORL, particularly impressive generalization to OOD preferences.

## 2. Related Work

**Offline MORL**   Most MORL research primarily focuses on online settings, aiming to train a general preference-conditioned policy (Hung et al., 2022; Lu et al., 2023a) or a set of single-objective policies (Natarajan & Tadepalli, 2005; Xu et al., 2020) that can generalize to arbitrary preferences. This paper considers an offline MORL setting that learns policy from static datasets without online interactions. PEDI (Wu et al., 2021) combines the dual gradient ascent with pessimism to multi-objective tabular MDPs. MO-SPIBB (Thomas et al., 2021) achieves safe policy improvement within a predefined set of policies under constraints. These methods often require a priori knowledge about target preferences. PEDA (Zhu et al., 2023a) first develops D4MORL benchmarks for scalable Offline MORL with a general preference for high-dimensional MDPs with continuous states and actions. It also attempts to solve the

Offline MORL via supervised learning, expanding return-conditioned sequential modeling to the MORL, such as MORvS (Emmons et al., 2021), and MODT (Chen et al., 2021). Lin et al. (2024) integrates policy-regularized into the MORL methods to alleviate the preference-inconsistent demonstration problem. This paper investigates the generalization problem of OOD preference, aiming to achieve better generalization through more expressive diffusion planners and the guidance of transition directions by a slider adapter.

**Diffusion Models for Decision Making**  Diffusion models (DMs) (Ho et al., 2020; Song et al., 2021b) have become a mainstream class of generative models. Their remarkable capabilities in complex distribution modeling demonstrate outstanding performance across various domains (Kingma et al., 2021; Ruiz et al., 2023), inspiring research to apply DMs to decision-making tasks (Chi et al., 2023; Yang et al., 2024; Xian et al., 2023). Applying DMs for decision-making tasks can mainly be divided into three categories (Zhu et al., 2023b): generating long horizon trajectories like planner (Janner et al., 2022; Dong et al., 2024a; Ajay et al., 2023), serving as more expressive multimodal polices (Wang et al., 2023; Hansen-Estruch et al., 2023; Yuan et al., 2023) and performing as the synthesizers for data augmentation (Lu et al., 2024). Diffusion models exhibit excellent generalization and distribution matching abilities (Kadkhodaie et al., 2024; Li et al., 2024b), and this property has also been applied to various types of decision-making tasks such as multi-task RL (He et al., 2024), meta RL (Ni et al., 2023) and aligning human feedback (Dong et al., 2023). In this paper, we further utilize sliding guidance to better stimulate the generalization capability of DMs, rather than merely fitting the distribution by data memorization (Gu et al., 2023).

## 3. Preliminaries

**Multi-objective RL**  The MORL problem can be formulated as a multi-objective Markov decision process (MOMDP) (Chatterjee et al., 2006). A MOMDP with $n$ objectives can be represented by a tuple $\langle \mathcal{S}, \mathcal{A}, \mathcal{P}, \mathcal{R}, \Omega, \gamma \rangle$, where $\mathcal{S}$ and $\mathcal{A}$ demote the state and action spaces, $\mathcal{P}$ is the transition function, $\mathcal{R} : \mathcal{S} \times \mathcal{A} \to \mathbb{R}^n$ is the reward function that outputs $n$-dim vector rewards $\boldsymbol{r} = \mathcal{R}(\boldsymbol{s}, \boldsymbol{a})$, $\Omega$ is the preference space containing vector preferences $\boldsymbol{\omega}$, and $\gamma$ is a discount factor. The policy $\pi$ is evaluated for $m$ distinct preferences $\{\boldsymbol{\omega}_p\}_{p=1}^m$, resulting policy set be represented as $\{\pi_p\}_{p=1}^m$, where $\pi_p = \pi(\boldsymbol{a}|\boldsymbol{s}, \boldsymbol{\omega}_p)$, and $\boldsymbol{G}^{\pi_p} = \mathbb{E}_{\boldsymbol{a}_t \sim \pi_p} [\sum_t \gamma^t \mathcal{R}(\boldsymbol{s}_t, \boldsymbol{a}_t)]$ is the corresponding unweighted expected return. We say the solution $\boldsymbol{G}^{\pi_p}$ is *dominated* by $\boldsymbol{G}^{\pi_q}$ when $G_i^{\pi_p} < G_i^{\pi_q}$ for $\forall i \in [1, 2, \cdots, n]$. The *Pareto front $P$* of the policy $\pi$ contains all its solutions that are not dominated. In MORL, we aim to define a policy such that its empirical Pareto front is a good approximation

of the true one. While not knowing the true front, we can define a set of metrics for relative comparisons among algorithms, e.g., *hypervolume*, *sparsity*, and *return deviation*, to which we give a further explanation in Section 5.

**Denoising Diffusion Implicit Models (DDIM)**  Assume the random variable $\boldsymbol{x}^0 \in \mathbb{R}^D$ follows an unknown distribution $q_0(\boldsymbol{x}^0)$. DMs define a *forward process* $\{\boldsymbol{x}^t\}_{t \in [0,T]}$[1] by the *noise schedule* $\{\alpha_t, \sigma_t\}_{t \in [0,T]}$, s.t., $\forall t \in [0, T]$:

$$\mathrm{d}\boldsymbol{x}^t = f(t)\boldsymbol{x}^t\mathrm{d}t + g(t)\mathrm{d}\boldsymbol{w}^t, \ \boldsymbol{x}^0 \sim q_0(\boldsymbol{x}^0), \quad (1)$$

where $\boldsymbol{w}^t \in \mathbb{R}^D$ is the standard Wiener process, and $f(t) = \frac{\mathrm{d}\log\alpha_t}{\mathrm{d}t}$, $g^2(t) = \frac{\mathrm{d}\sigma_t^2}{\mathrm{d}t} - 2\sigma_t^2\frac{\mathrm{d}\log\alpha_t}{\mathrm{d}t}$. The SDE forward process in Equation (1) has an *probability flow* ODE (PF-ODE) *reverse process* from time $T$ to 0 (Song et al., 2021b):

$$\frac{\mathrm{d}\boldsymbol{x}^t}{\mathrm{d}t} = f(t)\boldsymbol{x}^t - \frac{1}{2}g^2(t)\nabla_{\boldsymbol{x}}\log q_t(\boldsymbol{x}^t), \ \boldsymbol{x}^T \sim q_T(\boldsymbol{x}^T),$$
$$(2)$$

where the *score function* $\nabla_{\boldsymbol{x}}\log q_t(\boldsymbol{x}^t)$ is the only unknown term. Once we have estimated it, we can sample from $q_0(\boldsymbol{x}^0)$ by solving Equation (2). In practice, we train a neural network $\boldsymbol{\epsilon}_\theta(\boldsymbol{x}^t, t)$ to estimate the scaled score function $-\sigma_t\nabla_{\boldsymbol{x}}\log q_t(\boldsymbol{x}^t)$ by minimizing the score matching loss:

$$\mathcal{L}(\theta) = \mathbb{E}_{t \sim U(0,T), \boldsymbol{x}^0 \sim q_0(\boldsymbol{x}^0), \boldsymbol{\epsilon} \sim \mathcal{N}(\boldsymbol{0}, \boldsymbol{I})} \left[ \|\boldsymbol{\epsilon}_\theta(\boldsymbol{x}^t, t) - \boldsymbol{\epsilon}\|_2^2 \right].$$
$$(3)$$

Since $\boldsymbol{\epsilon}_\theta(\boldsymbol{x}^t, t)$ can be considered as a predicted Gaussian noise added to $\boldsymbol{x}^t$, it is called the *noise prediction model*. With a well-train noise prediction model, DDIM (Song et al., 2021a) discretizes the PF-ODE to the first order for solving, i.e., for each sampling step:

$$\boldsymbol{x}^t = \alpha_t \left( \frac{\boldsymbol{x}^s - \sigma_t\boldsymbol{\epsilon}_\theta(\boldsymbol{x}^s, s)}{\alpha_s} \right) + \sigma_s\boldsymbol{\epsilon}_\theta(\boldsymbol{x}^s, s) \quad (4)$$

holds for any $0 \le t < s \le T$. To enable the diffusion model to generate trajectories based on specified properties, Classifier Guidance (CG) (Dhariwal & Nichol, 2021) and Classifier-free Guidance (CFG) (Ho & Salimans, 2022) are the two main techniques. We use CFG due to its stable and easy-to-train properties. CFG directly uses a conditional noise predictor to guide the solver:

$$\tilde{\boldsymbol{\epsilon}}_\theta(\boldsymbol{x}^t, t, \boldsymbol{c}) := w \cdot \boldsymbol{\epsilon}_\theta(\boldsymbol{x}^t, t, \boldsymbol{c}) + (1 - w) \cdot \boldsymbol{\epsilon}_\theta(\boldsymbol{x}^t, t), \ (5)$$

where $w$ is used to control the guidance strength.

**Guided Sampling Methods**  To enable the diffusion model to generate trajectories based on specified properties, Classifier Guidance (CG) (Dhariwal & Nichol, 2021)

---

[1]To ensure clarity, we establish the convention that the superscript $t$ denotes the diffusion timestep, while the subscript $t$ represents the decision-making timestep.

and Classifier-free Guidance (CFG) (Ho & Salimans, 2022) are the two main techniques. CG requires training an additional classifier $\log p_\phi(c|x^t, t)$ to predict the log probability of exhibiting property $c$. During inference, the gradient of the classifier is used to guide the solver:

$$\bar{\epsilon}_\theta(x^t, t, c) = \epsilon_\theta(x^t, t) - w \cdot \sigma_t \nabla_x \mathcal{C}_\phi(x^t, t, c), \quad (6)$$

on the contrary, CFG directly uses a conditional noise predictor to guide the solver:

$$\tilde{\epsilon}_\theta(x^t, t, c) := w \cdot \epsilon_\theta(x^t, t, c) + (1 - w) \cdot \epsilon_\theta(x^t, t), \quad (7)$$

where $w$ is used to control the guidance strength.

## 4. Methodology

To address the challenges of expressiveness and generalization in Offline MORL, we propose a novel Offline MORL framework called MODULI. This framework utilizes conditional diffusion models for preference-oriented trajectory generation and establishes a general policy suitable for a wide range of preferences while also generalizing to OOD preferences. MODULI first defines a conditional generation and planning framework and corresponding preference-return normalization methods. Moreover, to enhance generalization capability, we design a sliding guidance mechanism, where a slider adapter actively adjusts the data distribution from known ID to specific OOD preference, thereby avoiding simple memorization.

### 4.1. Offline MORL as Conditional Generative Planning

We derive policies on the Offline MORL dataset $\mathcal{D}$ with trajectories $\tau = (\omega, s_0, a_0, r_0, \cdots, s_{T-1}, a_{T-1}, r_{T-1})$ to respond to arbitrary target preferences $\omega \in \Omega$ during the deployment phase. Previous Offline MORL (Zhu et al., 2023a) algorithms generally face poor expressiveness, unable to adapt to a wide range of preferences through a general preference-conditioned policy, resulting in sub-optimal approximate Pareto front. Inspired by the excellent expressiveness of diffusion models in decision making (Ajay et al., 2023; Dong et al., 2023), MODULI introduces diffusion models to model the policy. We define the sequential decision-making problem as a conditional generative objective $\max_\theta \mathbb{E}_{\tau \sim \mathcal{D}} \left[ \log p_\theta \left( x^0(\tau) \mid y(\tau) \right) \right]$, where $\theta$ is the trainable parameter. The diffusion planning in MORL aims to generate sequential states $x^0(\tau) = [s_0, \cdots, s_{H-1}]$ with horizon $H$ to satisfy the denoising conditions $y = [\omega, g]$, where $\omega$ is a specific preference. Similar to Zhu et al. (2023a), $g_t$ is defined as the vector-valued Returns-to-Go (RTG) for a single step, i.e. $g_t = \sum_t^T \mathcal{R}(s_t, a_t)$. The trajectory RTG $g$ is the average of the RTGs at each step.

**Training** Next, we can sample batches of $x^0(\tau)$ and $y$ from the dataset and update $p_\theta$ as the modified score matching objective in Equation (3):

$$\mathcal{L}(\theta) = \mathbb{E}_{(x^0, \omega, g) \sim \mathcal{D}, t \sim U(0, T), \epsilon \sim \mathcal{N}(0, I)} \left[ \| \epsilon_\theta(x^t, t, \omega, g) - \epsilon \|_2^2 \right]. \quad (8)$$

MODULI uses the CFG guidance for conditional generation. During training, MODULI learns both a conditioned noise predictor $\epsilon_\theta(x^t, t, \omega, g)$ and an unconditioned one $\epsilon_\theta(x^t, t, \varnothing)$. We adopt a masking mechanism for training, with a probability of $0 < p < 1$ to zero out the condition of a trajectory, equivalent to an unconditional case. We employ DiT1d (Dong et al., 2024b) network architecture instead of UNet (Ronneberger et al., 2015), which has been proven to have more precise generation capabilities. MODULI also employs a loss-weight trick, setting a higher weight for the next state $s_1$ of the trajectory $x^0(\tau)$ to encourage more focus on it, closely related to action execution. For details on the loss-weight trick, please refer to Appendix F.

### 4.2. Refining Guidance via Preference-Return Normalization Methods

**Global Normalization** To perform generation under different scales and physical quantities of objectives, it is essential to establish appropriate guiding conditions, making return normalization a critical component. We first consider directly extending the single-objective normalization scheme to the multi-objective case, named *Global Normalization*. We normalize each objective's RTG using min-max style, scaling them to a unified range. Specifically, we calculate the global maximum RTG $g^{\max} = [g_1^{\max}, \cdots, g_n^{\max}]$ for each objective in the offline dataset. The same applies to the minimum RTG $g^{\min}$. The conditioned RTG $g$ are first normalized to $(g - g^{\min})/(g^{\max} - g^{\min}) \in [0, 1]^n$, then concatenated with the preference $\omega$ as inputs. During deployment, we aim to generate $x(\tau)$ to align the given preference $\omega_{\text{target}}$ and maximized scalarized return using the highest RTG condition $g = 1^n$. Note that we compute the global extremum values based on the offline dataset with different preferences, which are not necessarily the reachable min/max value for each objective. Also, some objectives are inherently conflicting and can't maximized at the same time. As a result, direct guidance using the target conditions of $y = [\omega_{\text{target}}, 1^n]$ is unachievable for some preferences, leading to a performance drop. To address these issues, we match preferences with vector returns and then propose two novel normalization methods that adapt to multi-objective scenarios to achieve preference-related return normalization, providing more refined guidance: *Preference Predicted Normalization* and *Neighborhood Preference Normalization*:

**Preference Predicted Normalization** To avoid the issue in *Global Normalization*, we hope to obtain corresponding RTG conditioned on the given preference. The dataset contains a discrete preference space and it is not possi-

ble to directly obtain the maximum RTG for each preference. Therefore, we propose *Preference Predicted Normalization* training a generalized return predictor additionally $g^{\max}(\omega) = R_\psi(\omega)$ to capture the maximum expected return achievable for a given preference $\omega$. Specifically, we first identify all undominated solution trajectories $\tau_P \in D_P$, where $D_P \subseteq D$ and $P$ denotes *Pareto front*. Then the generalized return predictor is trained by minimizing the return prediction error conditioned on the preference $\omega$:

$$\mathcal{L}(\psi) = \mathbb{E}_{(\omega, \hat{g}) \sim D_P} \left[ \| R_\psi(\omega) - \hat{g} \|_2^2 \right], \qquad (9)$$

Then we calculate normalized RTG for each trajectory $\tau$ with preference $\omega$ as $g_\tau / g_{\max}(\omega) \in [0,1]^n$. With RTG prediction, we can follow closer to the training distribution, guiding the generation by specifying different maximum vector returns for each preference.

**Neighborhood Preference Normalization** We train $R_\psi(\omega)$ using only non-dominated solutions to predict the expected maximum RTG. If the quality of the behavior policy in the dataset is relatively low, the non-dominated solutions in the dataset are sparse, which introduces prediction errors that result in inaccurate normalization. Therefore we propose a simpler, learning-free method named *Neighborhood Preference Normalization*. In a linear continuous preference space, similar preferences typically lead to similar behaviors. Therefore, we can use a neighborhood set of trajectories to obtain extremum values, avoiding inaccurate prediction. For a trajectory $\tau$ and its corresponding preference $\omega_\tau$, we define $\epsilon$-neighborhood:

$$\mathcal{B}_\epsilon(\tau) = \{\tau' \mid \|\omega_{\tau'} - \omega_\tau\| \leq \epsilon\} \qquad (10)$$

Therefore, we can use the maximum expected return within the neighborhood trajectories to approximate the maximum expected return, normalized as follows:

$$g_\tau = \frac{g_\tau - \min\left(g_{\mathcal{B}_\epsilon(\tau)}\right)}{\max\left(g_{\mathcal{B}_\epsilon(\tau)}\right) - \min\left(g_{\mathcal{B}_\epsilon(\tau)}\right)} \in [0,1], \quad (11)$$

$$g_{\mathcal{B}_\epsilon(\tau)} = \{g_{\tau'} \mid \tau' \in \mathcal{B}_\epsilon(\tau)\}. \qquad (12)$$

Overall, we empirically find that the NPN method achieves consistently good performance across datasets of varying quality, making it the preferred method. On the other hand, PPN, which requires training an NN to estimate the return, performs well only in high-quality datasets. We provide depth analysis of the three normalization methods in the Section 5.3 and Appendix C

**Action Extraction** Now, MODULI can accurately generate desired trajectories with arbitrary preference and RTG in the CFG manner according to Equation (7). Then, we train an additional inverse dynamics model $a_t = h(s_t, s_{t+1})$ to

extract the action $a_t$ to be executed from generated $x$. It is worth noting that we fixed the initial state as $x^k[0] = x^0[0]$ to enhance the consistency of the generated results.

### 4.3. Enhancing OOD Preference Generalization with Sliding Guidance

After using diffusion models with appropriate normalization, MODULI can exhibit remarkable expressive capabilities, fully covering the ID distribution in the dataset. However, when faced with OOD preferences, it tends to conservatively generate the trajectory closest to the ID preferences. This may indicate that the diffusion model has overfitted the dataset rather than learning the underlying distribution. To enhance generalization to OOD preferences, inspired by Gandikota et al. (2024) and Du et al. (2020), we propose a novel sliding guidance mechanism. After training the diffusion models $p_\theta$, We additionally trained a plug-and-play slider adapter with the same network structure of diffusion models, which learns the latent direction of preference changes. This approach allows for greater control during generation by actively adjusting the target distribution, thereby preventing simple memorization. We define the adapter as $p_{\theta^*}$. When conditioned on $c$, this method boosts the possibility of attribute $c_+$ while reduces the possibility of attribute $c_-$ according to original model $p_\theta$:

$$p_\theta\left(x^0(\tau) \mid c\right) \leftarrow p_\theta\left(x^0(\tau) \mid c\right) \left(\frac{p_\theta^*\left(c_+ \mid x^0(\tau)\right)}{p_\theta^*\left(c_- \mid x^0(\tau)\right)}\right)^\eta \tag{13}$$

First, we train a diffusion model $p$ to generate trajectories corresponding to ID preferences. Then, we attempt to learn the pattern of change (for example, when the preference shifts from energy efficiency to high speed, the amplitude of the Swimmer's movements gradually increases). Therefore, our goal is to learn an adapter $p^*$, which applies this pattern to OOD preferences to achieve better generalization. In Equation (13), the exponential term represents increasing the likelihood of preference $c+$ and decreasing the likelihood of preference $c-$. $\eta$ represents the guidance strength weight in classifier-free guidance for the diffusion model. Based on Equation (13), we can use an adapter-style method to adjust the original noise prediction model ($p$). Now, the noise prediction combines the noise from the original model and the adapter. Then, we can derive a direct fine-tuning scheme that is equivalent to modifying the noise prediction model. The derivation process is provided in the Appendix D:

$$\epsilon_{\theta^*}\left(x^t, c, t\right) \leftarrow \epsilon_\theta^*\left(x^t, c, t\right) + \eta\left(\epsilon_\theta\left(x^t, c_+, t\right) - \epsilon_\theta\left(x^t, c_-, t\right)\right) \tag{14}$$

The score function proposed in Equation (14) justifies the distribution of the condition $c$ by converging towards the positive direction $c_+$ while diverging from the negative di-

rection $c_-$. Therefore, we can let the $\theta^*$ model capture the direction of preference changes, that is, we hope that the denoising process of $\epsilon_\theta^*(x^t, c, t)$ at each step meets the unit length shift, i.e. $\epsilon_{\theta^*}(x^t, c, t) = \frac{[\epsilon_\theta(x^t, c_+, t) - \epsilon_\theta(x^t, c_-, t)]}{c_+ - c_-}$. In MORL tasks, the condition $c$ is $\omega$, and $c^+$ and $c^-$ are the changes in preference $\Delta\omega$ in the positive and negative directions, respectively. We minimize the loss $\mathcal{L}(\theta^*)$ to train the adapter:

$$\mathcal{L}(\theta^*) = \mathbb{E}_{(\boldsymbol{\omega}, \Delta\boldsymbol{\omega}) \sim D} \left[ \left\| \epsilon_{\theta^*}(\boldsymbol{x}^t, \boldsymbol{\omega}, t) - \frac{[\epsilon_\theta(\boldsymbol{x}^t, \boldsymbol{\omega} + \Delta\boldsymbol{\omega}, t) - \epsilon_\theta(\boldsymbol{x}^t, \boldsymbol{\omega} - \Delta\boldsymbol{\omega}, t)]}{2\Delta\boldsymbol{\omega}} \right\|_2^2 \right] \tag{15}$$

During deployment, when we encounter OOD preferences $\boldsymbol{\omega}_{\text{OOD}}$, we can first transition from the nearest ID preference $\boldsymbol{\omega}_{\text{ID}}$. Then, we calculate $\Delta\boldsymbol{\omega} = \boldsymbol{\omega}_{\text{OOD}} - \boldsymbol{\omega}_{\text{ID}}$, and at each denoising step, simultaneously apply the diffusion model and adapter as $\epsilon_\theta + \Delta\boldsymbol{\omega}\epsilon_{\theta^*}$, so that the trajectory distribution gradually shifts from ID preferences to OOD preferences to achieve better generalization. We provide the pseudo code for the entire training and deployment process in Appendix E.

## 5. Experiments

We conduct experiments on various Offline MORL tasks to study the following research questions (RQs): **Expressiveness (RQ1)** How does MODULI perform on `Complete` dataset compared to other baselines? **Generalizability (RQ2)** Does MODULI exhibit leading generalization performance in the OOD preference of `Shattered` or `Narrow` datasets? **Normalization (RQ3)** How do different normalization methods affect performance? All experimental details of MODULI are in Appendix B.

**Datasets and Baselines** We evaluate MODULI on D4MORL (Zhu et al., 2023a), an Offline MORL benchmark consisting of offline trajectories from 6 multi-objective MuJoCo environments. D4MORL contains two types of quality datasets: `Expert` dataset collected by pre-trained expert behavioral policies, and `Amateur` dataset collected by perturbed behavioral policies. We named the original dataset of D4MORL as the `Complete` dataset, which is used to verify expressiveness ability to handle multi-objective tasks. Additionally, to evaluate the algorithm's generalization ability to out-of-distribution (OOD) preferences, similar to Figure 1, we collected two types of preference-deficient datasets, namely `Shattered` and `Narrow`. We refer to the Appendix A.2 for the dataset generation.

- `Shattered`: Simulates incomplete preference distribution by removing part of the trajectories from the Complete dataset, creating regions with missing preferences. It primarily evaluates the interpolation OOD generalization ability, assessing how well the algorithm generalizes when encountering unseen preferences with some missing.

- `Narrow`: Similar to the Shattered dataset, but it mainly evaluates the extrapolation OOD generalization ability. The Narrow dataset is generated by removing a portion of the trajectories from both ends of the Complete dataset, resulting in a narrower preference distribution.

We compare MODULI with various strong offline MORL baselines: BC(P), MORvS(P) (Emmons et al., 2021), MODT(P) (Chen et al., 2021) and MOCQL(P) (Kumar et al., 2020). These algorithms are modified variants of their single-objective counterparts and well-studied in PEDA (Zhu et al., 2023a). MODT(P) and MORvS(P) view MORL as a preference-conditioned sequence modeling problem with different architectures. BC(P) performs imitation learning through supervised loss, while MOCQL(P) trains a preference-conditioned Q-function based on temporal difference learning. Further details of datasets and baselines are described in Appendix A.

**Metrics** We use two commonly used metrics **HV** and **SP** for evaluating the empirical Pareto front and an additional metric **RD** for evaluating the generalization of OOD preference regions: ❶ Hypervolume (**HV**) $\mathcal{H}(P)$ measures the space enclosed by solutions in the Pareto front $P$: $\mathcal{H}(P) = \int_{\mathbb{R}^n} \mathbf{1}_{H(P)}(\boldsymbol{z}) \mathrm{d}\boldsymbol{z}$, where $H(P) = \{\boldsymbol{z} \in \mathbb{R}^n | \exists i : 1 \leq i \leq |P|, \boldsymbol{r_0} \preceq \boldsymbol{z} \preceq P(i)\}$, $\boldsymbol{r_0}$ is a reference point determined by the environment, $\preceq$ is the dominance relation operator and $\mathbf{1}_{H(P)}$ equals 1 if $\boldsymbol{z} \in H(P)$ and 0 otherwise. Higher HV is better, implying empirical Pareto front expansion. ❷ Sparsity (**SP**) $\mathcal{S}(P)$ measures the density of the Pareto front $P$: $\mathcal{S}(P) = \frac{1}{|P|-1} \sum_{i=1}^n \sum_{k=1}^{|P|-1} (\tilde{P}_i(k) - \tilde{P}_i(k+1))^2$, where $\tilde{P}_i$ represents the sorted list of values of the $i$-th target in $P$ and $\tilde{P}_i(k)$ is the $k$-th value in $\tilde{P}_i$. Lower SP, indicating a denser approximation of the Pareto front, is better. ❸ Return Deviation (**RD**) $\mathcal{R}(O)$ measures the discrepancy between the solution set $O$ in the out-of-distribution preference region and the maximum predicted value: $\mathcal{R}(O) = \frac{1}{|O|} \sum_{k=1}^{|O|} \|O(k) - \boldsymbol{R}_\psi(\boldsymbol{\omega}_{O(k)})\|_2^2$, where $\boldsymbol{R}_\psi$ is pretrained generalized return predictor mentioned in Section 4.2, $\boldsymbol{\omega}_{O(k)}$ is the corresponding preference of $O(k)$. Lower RD is better, as it indicates that the performance of the solution is closer to the predicted maximum under OOD preference, suggesting better generalization ability. Following the Zhu et al. (2023a), we uniformly select 501 preference points for $2obj$ environments and 325 points for $3obj$ environments. For each experiment, we use 3 random seeds and report the mean and standard deviation.

### 5.1. Evaluating Expressiveness on `Complete` Datasets

We first compare MODULI with various baselines on the `Complete` datasets. In this dataset version, the trajectories have comprehensive coverage preferences, and the policy needs sufficient expressive capability to cover complex dis-

*Table 1.* HV and SP performance on full `Complete` datasets. (B: behavioral policy). The best scores are emphasized in bold.

| | Env | Metrics | B | MODT(P) | MORvS(P) | BC(P) | CQL(P) | MODULI |
|---|---|---|---|---|---|---|---|---|
| Amateur | Ant | HV ($\times 10^6$) ↑ | 5.61 | 5.92±.04 | 6.07±.02 | 4.37±.06 | 5.62±.23 | **6.08±.03** |
| | | SP ($\times 10^4$) ↓ | - | 8.72±.77 | 5.24±.52 | 25.90±16.40 | 1.06±.28 | **0.53±.05** |
| | HalfCheetah | HV ($\times 10^6$) ↑ | 5.68 | 5.69±.01 | **5.77±.00** | 5.46±.02 | 5.54±.02 | 5.76±.00 |
| | | SP ($\times 10^4$) ↓ | - | 1.16±.42 | 0.57±.09 | 2.22±.91 | 0.45±.27 | **0.07±.02** |
| | Hopper | HV ($\times 10^7$) ↑ | 1.97 | 1.81±.05 | 1.76±.03 | 1.35±.03 | 1.64±.01 | **2.01±.01** |
| | | SP ($\times 10^5$) ↓ | - | 1.61±.29 | 3.50±1.54 | 2.42±1.08 | 3.30±5.25 | **0.10±.01** |
| | Swimmer | HV ($\times 10^4$) ↑ | 2.11 | 1.67±.22 | 2.79±.03 | 2.82±.04 | 1.69±.93 | **3.20±.00** |
| | | SP ($\times 10^0$) ↓ | - | 2.87±1.32 | **1.03±.20** | 5.05±1.82 | 8.87±6.24 | 9.50±.59 |
| | Walker2d | HV ($\times 10^6$) ↑ | 4.99 | 3.10±.34 | 4.98±.01 | 3.42±.42 | 1.78±.33 | **5.06±.00** |
| | | SP ($\times 10^4$) ↓ | - | 164.20±13.50 | 1.94±.06 | 53.10±34.60 | 7.33±5.89 | **0.25±.03** |
| | Hopper-3obj | HV ($\times 10^{10}$) ↑ | 3.09 | 1.04±.16 | 2.77±.24 | 2.42±.18 | 0.59±.42 | **3.33±.06** |
| | | SP ($\times 10^5$) ↓ | - | 10.23±2.78 | 1.03±.11 | 0.87±.29 | 2.00±1.72 | **0.10±.00** |
| Expert | Ant | HV ($\times 10^6$) ↑ | 6.32 | 6.21±.01 | 6.36±.02 | 4.88±.17 | 5.76±.10 | **6.39±.02** |
| | | SP ($\times 10^4$) ↓ | - | 8.26±2.22 | 0.87±.19 | 46.20±16.40 | 0.58±.10 | **0.79±.12** |
| | HalfCheetah | HV ($\times 10^6$) ↑ | 5.79 | 5.73±.00 | 5.78±.00 | 5.54±.05 | 5.63±.04 | **5.79±.00** |
| | | SP ($\times 10^4$) ↓ | - | 1.24±.23 | 0.67±.05 | 1.78±.39 | 0.10±.00 | **0.07±.00** |
| | Hopper | HV ($\times 10^7$) ↑ | 2.09 | 2.00±.02 | 2.02±.02 | 1.23±.10 | 0.33±.39 | **2.09±.01** |
| | | SP ($\times 10^5$) ↓ | - | 16.30±10.60 | 3.03±.36 | 52.50±4.88 | 2.84±2.46 | **0.09±.01** |
| | Swimmer | HV ($\times 10^4$) ↑ | 3.25 | 3.15±.02 | **3.24±.00** | 3.21±.00 | 3.22±.08 | **3.24±.00** |
| | | SP ($\times 10^0$) ↓ | - | 15.00±7.49 | **4.39±.07** | 4.50±.39 | 13.60±5.31 | 4.43±.38 |
| | Walker2d | HV ($\times 10^6$) ↑ | 5.21 | 4.89±.05 | 5.14±.01 | 3.74±.11 | 3.21±.32 | **5.20±.00** |
| | | SP ($\times 10^4$) ↓ | - | 0.99±.44 | 3.22±.73 | 75.60±52.30 | 6.23±10.70 | **0.11±.01** |
| | Hopper-3obj | HV ($\times 10^{10}$) ↑ | 3.73 | 3.38±.05 | 3.42±.10 | 2.29±.07 | 0.78±.24 | **3.57±.02** |
| | | SP ($\times 10^5$) ↓ | - | 1.40±.44 | 2.72±1.93 | 0.72±.09 | 2.60±3.14 | **0.07±.00** |

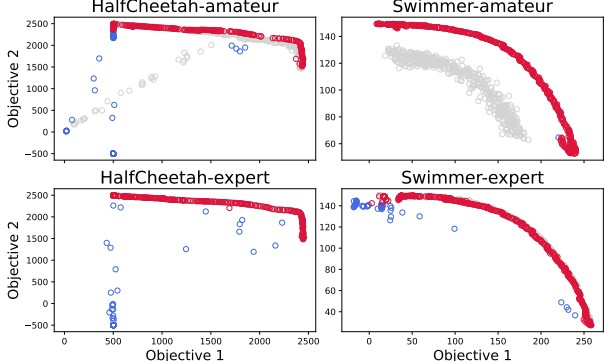

*Figure 2.* Approximated Pareto fronts on `Complete` dataset by MODULI. Undominated / Dominated solutions are colored in red / blue, and dataset trajectories are colored in grey.

tributions under a wide range of preferences. As shown in Table 1, most of the baselines (MOCQL, MOBC, MODT) exhibit sub-optimal performance, fail to achieve HV close to behavioral policies, and are prone to sub-preference space collapses that do not allow for dense solutions. MORvS can exhibit better HV and SP but fails to achieve performance

competitive with behavioral policies due to the poor expressiveness of the context policy. Overall, MODULI achieves the most advanced policy performance according to HV, and the most dense solution with the lowest SP. We visualize the empirical Pareto front in Figure 2, demonstrating that MODULI is a good approximator under different data qualities. In the expert dataset, MODULI accurately replicates the behavioral policies. In the amateur dataset, due to the inclusion of many suboptimal solutions, MODULI significantly expands the Pareto front, demonstrating its ability to stitch and guide the generation of high-quality trajectories.

### 5.2. Evaluating Generalization on `Shattered` and `Narrow` Datasets

**Performance** We evaluate the generalization performance on the `Shattered` and `Narrow` datasets. Table 2 shows the comparison results on the expert datasets, where MODULI still demonstrates the best performance across all baselines. Specifically, MODULI achieves significant improvements on the new RD metric, indicating better generalization capability and a better approximation of the Pareto front in OOD preference regions. We further visualize the powerful OOD preference generalization ability of MODULI.

*Table 2.* HV, SP and RD on `Shattered`/`Narrow` datasets. The best scores are emphasized in bold. See Appendix G for full results.

| Env | Metrics | Shattered | | | Narrow | | |
|---|---|---|---|---|---|---|---|
| | | **MODT(P)** | **MORvS(P)** | **MODULI** | **MODT(P)** | **MORvS(P)** | **MODULI** |
| Ant | HV ($\times 10^6$) ↑ | 5.88±.02 | 6.38±.01 | **6.39±.01** | 5.05±.08 | 6.06±.00 | **6.36±.00** |
| | RD ($\times 10^3$) ↓ | 0.71±.03 | 0.16±.00 | **0.14±.01** | 0.98±.01 | 0.37±.03 | **0.25±.01** |
| | SP ($\times 10^4$) ↓ | 2.22±.50 | 0.75±.13 | **0.53±.13** | 0.88±.41 | **0.80±.11** | **0.80±.25** |
| HalfCheetah | HV ($\times 10^6$) ↑ | 5.69±.00 | 5.73±.00 | **5.78±.00** | 5.04±.00 | 5.46±.01 | **5.76±.00** |
| | RD ($\times 10^3$) ↓ | 0.18±.00 | 0.10±.00 | **0.07±.00** | 0.43±.00 | 0.29±.00 | **0.17±.00** |
| | SP ($\times 10^4$) ↓ | 0.18±.01 | 0.15±.02 | **0.11±.02** | 0.04±.00 | 0.05±.00 | **0.04±.00** |
| Hopper | HV ($\times 10^7$) ↑ | 1.95±.03 | 2.06±.01 | **2.07±.00** | 1.85±.02 | 1.98±.00 | **2.04±.01** |
| | RD ($\times 10^3$) ↓ | 1.42±.00 | 1.10±.04 | **0.22±.06** | 4.24±.02 | **1.37±.05** | 2.42±.05 |
| | SP ($\times 10^5$) ↓ | 0.85±.27 | 0.20±.06 | **0.11±.01** | 0.34±.07 | **0.16±.02** | 0.25±.05 |
| Swimmer | HV ($\times 10^4$) ↑ | 3.20±.00 | **3.24±.00** | **3.24±.00** | 3.03±.00 | 3.10±.00 | **3.21±.00** |
| | RD ($\times 10^2$) ↓ | 0.38±.00 | 0.16±.00 | **0.06±.00** | 0.38±.00 | 0.39±.01 | **0.10±.00** |
| | SP ($\times 10^0$) ↓ | 15.50±.33 | 7.36±.81 | **5.79±.43** | 3.65±.22 | 4.38±.80 | **3.28±.15** |
| Walker2d | HV ($\times 10^6$) ↑ | 5.01±.01 | 5.14±.01 | **5.20±.00** | 4.75±.01 | 4.85±.01 | **5.10±.01** |
| | RD ($\times 10^3$) ↓ | 0.86±.01 | 0.65±.02 | **0.15±.01** | 1.07±.01 | 1.45±.02 | **0.28±.02** |
| | SP ($\times 10^4$) ↓ | 0.49±.03 | 0.27±.04 | **0.12±.01** | 0.34±.09 | 0.19±.02 | **0.13±.01** |
| Hopper-3obj | HV ($\times 10^{10}$) ↑ | 2.83±.06 | 3.28±.07 | **3.43±.02** | 3.18±.05 | 3.32±.00 | **3.37±.05** |
| | RD ($\times 10^3$) ↓ | 1.60±.01 | 1.60±.02 | **1.28±.03** | 2.48±.00 | **2.24±.02** | 2.38±.01 |
| | SP ($\times 10^5$) ↓ | 0.09±.01 | **0.06±.00** | 0.10±.01 | 0.11±.01 | 0.22±.02 | **0.12±.01** |

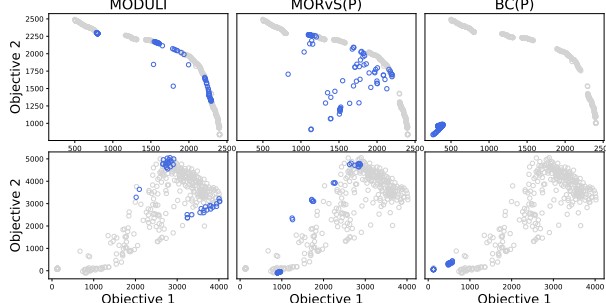

*Figure 3.* Solution (Blue points) under OOD preference for different algorithms in Walker2d-expert-Shattered (Top) and Hopper-amateur-Narrow (Bottom). Dataset trajectories are colored in grey.

**OOD Generalization Visualization** In Figure 3, we visualize the actual solutions obtained under OOD preferences. Gray points represent the trajectories in the dataset, where the preference of the dataset is incomplete. We observe that BC(P) exhibits policy collapse when facing OOD preferences, showing extremely low returns and a mismatch with the preferences. RvS(P) demonstrates limited generalization ability, but most solutions still do not meet the preference requirements. Only MODULI can produce correct solutions even when facing OOD preferences. In the `Shattered` dataset, MODULI can patch partially OOD preference regions, while in the `Narrow` dataset, MODULI can slightly extend the Pareto front to both sides.

**Ablation of Sliding Guidance** We conducted an ablation study of the sliding guidance mechanism on 9 tasks within `Narrow` datasets. The results are presented in the radar chart in Figure 4. We found that in most tasks, MODULI with slider exhibits higher HV and lower RD, indicating leading performance in MO tasks. At the same time, the presence or absence of the slider has little impact on SP performance, indicating that sliding guidance distribution does not impair the ability to express the ID preferences. Besides, we visualized the empirical Pareto front in Figure 5 for `Hopper-Narrow` dataset. The sliding bootstrap mechanism significantly extends the Pareto set of OOD preference regions, exhibiting better generalization. Due to space limits, we provide more comparative experiments on the different guidance mechanisms (CG, CG+CFG) in Appendix H.

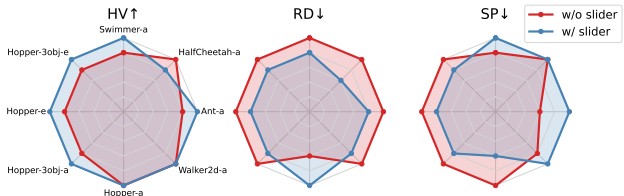

*Figure 4.* Comparison of HV, SP, and RD performance of MODULI with and without slider across various environments. All metrics are rescaled by dividing by their maximum values. The suffix "-e" denotes `Expert` and "-a" denotes `Amateur`.

### 5.3. Comparison of Normalization Methods

We compared the impact of different return regularization methods on MODULI's performance on the `Complete` dataset. As shown in Table 3, Neighborhood Preference Normalization (NP Norm) consistently has the highest HV performance. On the expert dataset, both NP Norm and

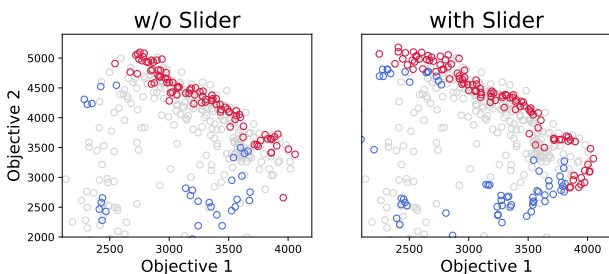

*Figure 5.* The Pareto front illustrations for MODULI with and without the Slider, generated on the Hopper-`Amateur-Narrow` dataset, demonstrate that the inclusion of the Slider significantly extends the boundary of the policies.

PP Norm exhibit very similar HV values, as they can accurately estimate the highest return corresponding to each preference, whether through prediction or neighborhood analysis. However, on the amateur dataset, PP Norm fails to predict accurately, resulting in lower performance. We also noted that the naive method of Global Norm fails to approximate the Pareto front well, but in some datasets, it achieved denser solutions (small SP). We believe this is because the Global Norm produced inaccurate and homogenized guiding information, leading to different preferences for similar solutions, and resulting in dense but low-quality solutions. We refer to Appendix C for in-depth visual analysis.

*Table 3.* Comparison of normalization on `Complete` datasets. (**NP**: Neighborhood Preference, **PP**: Preference Predicted)

| | Env | Metrics | NP Norm | PP Norm | Global Norm |
|---|---|---|---|---|---|
| Amateur | Hopper | HV ($\times 10^7$) | **1.98±.02** | 1.94±.01 | 1.58±.02 |
| | | SP ($\times 10^5$) | 0.21±.06 | 0.24±.06 | **0.12±.03** |
| | Walker2d | HV ($\times 10^6$) | **5.03±.00** | 4.94±.00 | 4.02±.01 |
| | | SP ($\times 10^4$) | 0.20±.01 | 0.31±.01 | **0.04±.01** |
| Expert | Hopper | HV ($\times 10^7$) | **2.07±.00** | **2.07±.01** | 1.48±.02 |
| | | SP ($\times 10^5$) | 0.14±.04 | **0.12±.02** | 2.69±1.97 |
| | Walker2d | HV ($\times 10^6$) | **5.20±.00** | 5.18±.00 | 2.70±.04 |
| | | SP ($\times 10^4$) | **0.11±.01** | 0.17±.01 | 5.11±6.88 |

## 6. Conclusion

In this paper, we propose MODULI, a diffusion planning framework with two multi-objective normalization schemes, enabling preference-return condition-guided trajectory generation for decision-making. To achieve better generalization in OOD preferences, MODULI includes a sliding guidance mechanism, training an additional slider adapter for active distribution adjustment. We conduct extensive experiments on *Complete*, *Shattered*, and *Narrow* datasets, demonstrating the superior expressive and generalization capabilities of MODULI. The results show that MODULI is an excellent Pareto front approximator, capable of expanding the Pareto front and obtaining advanced, dense solutions. However, there are limitations, such as experiments con-

ducted in low-dimensional state space and continuous action space. We hope to extend it to multi-objective tasks with image inputs, further enhancing its generalization ability. We also believe that integrating evolutionary reinforcement learning (Li et al., 2024c;a; Hao et al., 2023) is a possible direction for future extension. Additionally, nonlinear preference spaces may present new challenges.

## Impact Statement

This paper investigates Offline MORL algorithms, focusing on their potential to address real-world problems where simultaneous optimization of multiple conflicting objectives is required and data collection is limited or expensive. By advancing Offline MORL methods, this work contributes to safer, more efficient, and practical deployment of reinforcement learning in fields such as healthcare, autonomous driving, and resource management, where learning from offline data is crucial. The findings and methodologies presented in this chapter are expected to inspire further research and facilitate the adoption of MORL in complex decision-making scenarios.

## Acknowledge

This work is supported by the National Natural Science Foundation of China (Grant Nos. 62422605, 92370132). We would like to thank all the anonymous reviewers for their valuable comments and constructive suggestions, which have greatly improved the quality of this paper.

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

# A. Dataset and Baselines

## A.1. D4MORL Benchmark

In this paper, we use the Datasets for Multi-Objective Reinforcement Learning (D4MORL) (Zhu et al., 2023a) for experiments, a large-scale benchmark for Offline MORL. This benchmark consists of offline trajectories from 6 multi-objective MuJoCo environments, including 5 environments with 2 objectives each, namely MO-Ant, MO-HalfCheetah, MO-Hopper, MO-Swimmer, MO-Walker2d, and one environment with three objectives: MO-Hopper-3obj. The objectives in each environment are conflicting, such as high-speed running and energy saving. In D4MORL, Each environment uses PGMORL's (Xu et al., 2020) behavioral policies for data collection. PGMORL trains a set of policies using evolutionary algorithms to approximate the Pareto front, which can map to the closest preference in the preference set for any given preference. Therefore, in D4MORL, two different quality datasets are defined: the **Expert Dataset**, which is collected entirely using the best preference policies in PGMORL for decision making, and the **Amateur Dataset**, which has a probability $P$ of using perturbed preference policies and a probability $1 - p$ of being consistent with the expert dataset, with $p = 0.65$. We can simply consider the **Amateur Dataset** as a relatively lower quality dataset with random perturbations added to the **Expert Dataset**. Each trajectory in the D4MORL dataset is represented as: $\tau = [\boldsymbol{\omega}, \boldsymbol{s}_0, \boldsymbol{a}_0, \boldsymbol{r}_0, \cdots, \boldsymbol{s}_T, \boldsymbol{a}_T, \boldsymbol{r}_T]$, where $T$ is episode length.

## A.2. `Complete`, `Shattered` and `Narrow` Datasets

In this section, we provide a detailed description of how three different levels of datasets are collected, and what properties of the algorithm they are used to evaluate respectively:

- `Complete` datasets i.e. the original version of the D4MORL datasets, cover a wide range of preference distributions with fewer OOD preference regions. These datasets primarily evaluate the expressive ability of algorithms and require deriving the correct solution based on diverse preferences.

- `Shattered` datasets simulate the scenario where the preference distribution in the dataset is incomplete, resulting in areas with missing preferences. This kind of dataset primarily evaluates the interpolation OOD generalization ability of algorithms, aiming for strong generalization capability when encountering unseen preferences with some lacking preferences. Specifically, the `Shattered` dataset removes part of the trajectories from the `Complete` dataset, creating $n$ preference-missing regions, which account for $m\%$ of the trajectories. If the total number of trajectories in the dataset is $N$, then a total of $n_{\text{lack}} = N * m\%$ trajectories are removed. First, all trajectories in the dataset are sorted based on the first dimension of their preferences $\boldsymbol{\omega}$, then $n$ points are selected at equal intervals according to the sorting results. Centered on these $n$ points, an equal number of $n_{\text{lack}}/n$ trajectories around each point are uniformly deleted. In this paper, $m, n$ are fixed to 30, and 3, respectively.

- The role of `Narrow` datasets is similar to `Shattered` datasets but mainly evaluates the extrapolation OOD generalization ability of algorithms. If we need to remove a total of $m\%$ of the trajectories, we can obtain the `Narrow` dataset with a narrow preference distribution by removing the same amount, $m/2\%$, from both ends of the sorted `Complete` dataset. In this paper, $m$ is fixed to 30.

## A.3. Baselines

In this paper, we compare with various strong baselines: MODT(P) (Chen et al., 2021), MORvS(P) (Emmons et al., 2021), BC(P), and MOCQL(P) (Kumar et al., 2020), all of which are well-studied baselines under the PEDA (Zhu et al., 2023a), representing different paradigms of learning methods. All baselines were trained using the default hyperparameters and official code as PEDA. MODT(P) extends DT architectures to include preference tokens and vector-valued returns. Additionally, MODT(P) also concatenates $\boldsymbol{\omega}$ to state tokens $[\boldsymbol{s}, \boldsymbol{\omega}]$ and action tokens $[\boldsymbol{a}, \boldsymbol{\omega}]$ for training. MORvS(P) and MODT(P) are similar, extending the MLP structure to adapt to MORL tasks. MORvS(P) concatenates the preference with the states and the average RTGs by default as one single input. BC(P) serves as an imitation learning paradigm to train a mapping from states with preference to actions. BC(P) simply uses supervised loss and MLPs as models. MOCQL(P) represents the temporal difference learning paradigm, which learns a preference-conditioned Q-function. Then, MOCQL(P) uses underestimated Q functions for actor-critic learning, alleviating the OOD phenomenon in Offline RL.

## A.4. Evaluation Metrics

In MORL, our goal is to train a policy to approximate the true Pareto front. While we do not know the true Pareto front for many problems, we can define metrics for evaluating the performance of different algorithms. Here We define several metrics for evaluation based on the empirical Pareto front $P$, namely Hypervolume (HV), Sparsity (SP), and Return Deviation (RD), we provide formal definitions as below:

**Definition A.1** (Hypervolume (**HV**)). Hypervolume $\mathcal{H}(P)$ measures the space enclosed by the solutions in the Pareto front $P$:

$$\mathcal{H}(P) = \int_{\mathbb{R}^n} \mathbf{1}_{H(P)}(\boldsymbol{z}) \mathrm{d}\boldsymbol{z},$$

where $H(P) = \{\boldsymbol{z} \in \mathbb{R}^n | \exists i : 1 \leq i \leq |P|, \boldsymbol{r_0} \preceq \boldsymbol{z} \preceq P(i)\}$, $\boldsymbol{r_0}$ is a reference point determined by the environment, $\preceq$ is the dominance relation operator and $\mathbf{1}_{H(P)}$ equals 1 if $\boldsymbol{z} \in H(P)$ and 0 otherwise. Higher HV is better, implying empirical Pareto front expansion.

**Definition A.2** (Sparsity (**SP**)). Sparsity $\mathcal{S}(P)$ measures the density of the Pareto front $P$:

$$\mathcal{S}(P) = \frac{1}{|P| - 1} \sum_{i=1}^{n} \sum_{k=1}^{|P|-1} (\tilde{P}_i(k) - \tilde{P}_i(k+1))^2,$$

where $\tilde{P}_i$ represents the sorted list of values of the $i$-th target in $P$ and $\tilde{P}_i(k)$ is the $k$-th value in $\tilde{P}_i$. Lower SP, indicating a denser approximation of the Pareto front, is better.

**Definition A.3** (Return Deviation (**RD**)). The return deviation $\mathcal{R}(O)$ measures the discrepancy between the solution set $O$ in the out-of-distribution preference region and the maximum predicted value:

$$\mathcal{R}(O) = \frac{1}{|O|} \sum_{k=1}^{|O|} \|O(k) - \boldsymbol{R}_\psi(\boldsymbol{\omega}_{O(k)})\|_2^2$$

where $\boldsymbol{R}_\psi$ is a generalized return predictor to predict the maximum expected return achievable for a given preference $\boldsymbol{\omega}$. To accurately evaluate RD, we consistently train this network $\boldsymbol{R}_\psi$ using the `Complete-Expert` dataset, as shown in Equation (9). In this way, for any preference of a missing dataset, we can predict and know the oracle's solution. *Please note that $\boldsymbol{R}_\psi$ here is only used to evaluate algorithm performance and is unrelated to the Preference Predicted Normalization method in MODULI.* $\boldsymbol{\omega}_{O(k)}$ is the corresponding preference of $O(k)$. A lower RD is better, as it indicates that the performance of the solution is closer to the predicted maximum under OOD preference, suggesting better generalization ability.

## B. Implementation Details

In this section, we provide the implementation details of MODULI.

- We utilize a DiT structure similar to AlignDiff (Dong et al., 2023) as the backbone for all diffusion models, with an embedding dimension of 128, 6 attention heads, and 4 DiT blocks. We use a 2-layer MLP for condition embedding with inputs $[\boldsymbol{\omega}, \boldsymbol{g}]$. For Preference Prediction Normalization, we use a 3-layer MLP to predict the maximum expected RTG for given preferences. For Neighborhood Preference Normalization, The neighborhood hyperparameter $\epsilon$ is fixed at $1e-3$ for 2-obj experiments and $3e-3$ for 3-obj experiments. For the slider adapter, we use a DiT network that is the same as the diffusion backbone of the slider. We observe that using the Mish activation (Misra, 2019) is more stable in OOD preference regions compared to the ReLU activation, which tends to diverge quickly. Finally, the model size of MODULI is around 11M.

- All models utilize the AdamW (Loshchilov & Hutter, 2017) optimizer with a learning rate of $2e-4$ and weight decay of $1e-5$. We perform a total of 200K gradient updates with a batch size of 64 for all experiments. We employ the exponential moving average (EMA) model with an ema rate of 0.995.

- For all diffusion models, we use a continuous linear noise schedule (Nichol & Dhariwal, 2021) for $\alpha_s$ and $\sigma_s$. We employ DDIM (Song et al., 2021a) for denoising generation and use a sampling step of 10. For all tasks, we use a sampling temperature of 0.5.

- The length of the planning horizon $H$ of the diffusion model should be adequate to capture the agent's behavioral preferences. Therefore, MODULI uses different planning horizons for different environments according to their properties. For Hopper and Walker2d tasks, we use a planning horizon of 32; for other tasks, we use a planning horizon of 4. For conditional sampling, we use guidance weight $w$ as 1.5.

- The inverse dynamic model is implemented as a 3-layer MLP. The first two layers consist of a Linear layer followed by a Mish (Misra, 2019) activation and a LayerNorm. And, the final layer is followed by a Tanh activation. This model utilizes the AdamW optimizer and is trained for 500K gradient steps.

- For all 2-obj tasks, we evaluate the model on 501 preferences uniformly distributed in the preference space. For all 3-obj tasks, we evaluate the model on 325 preferences uniformly distributed in the preference space. We perform three rounds of evaluation with different seeds and report the mean and standard deviation. Please note that in the `Complete` dataset, we default to not using the slider adapter for sliding guidance because there is no OOD preference.

- We standardized the parameter size of all baseline algorithms to the same magnitude as much as possible. MORvS: 8.55M, MOBC:8.83M, MODT:9.75M.

**Compute Resources**  We conducted our experiments on an Intel(R) Xeon(R) Platinum 8171M CPU @ 2.60GHz processor based system. The system consists of 2 processors, each with 26 cores running at 2.60GHz (52 cores in total) with 32KB of L1, 1024 KB of L2, 40MB of unified L3 cache, and 250 GB of memory. Besides, we use 8 Nvidia RTX3090 GPUs to facilitate the training procedure. The operating system is Ubuntu 18.04.

## C. In-depth Analysis of Different Normalization Methods

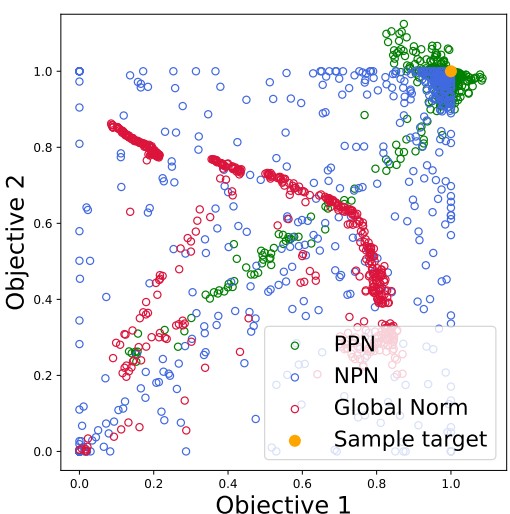

*Figure 6.* Visualization of trajectory returns for the Walker2d-amateur-Complete dataset after applying different normalization methods. (**PPN**: Preference Predicted Normalization, **NPN**: Neighborhood Preference Normalization, **Sample Target**: In the deployment phase, we fix the condition as $[\omega, 1^n]$ to guide the generation of high-quality trajectories.)

As shown in Figure 6, we visualize the trajectory returns of the Walker2d-amateur-Complete dataset after applying different normalization methods. In order to achieve refined guided generation, we require consistency between the training objectives and the sampling objectives, meaning the sampling objectives should correspond to high-quality trajectories in the dataset. A simple Global Normalization fails to achieve this, as it cannot identify a definitive high-return guiding objective that matches the return distribution of dataset. Setting the return to $1^n$ may be unachievable for Global Normalization. On the contrary, Preference Predicted Normalization (PPN) and Neighborhood Preference Normalization (NPN) successfully normalize the trajectories in the dataset into distinctly different trajectories. High-quality trajectories that conform to preferences are clustered around $1^n$, and trajectories points farther from $1^n$ represent lower vector returns. Note that at this time, the training objective and the sampling objective are fully aligned. Compared with NPN, the PPN method uses neural networks to predict the maximum expected RTG for each preference, facing prediction errors in low-quality datasets (e.g. amateur

dataset). After PPN process using $g_\tau/g_{\max}(\boldsymbol{\omega})$, a small number of trajectories even slightly exceed $\mathbf{1}^n$, indicating that $R_\psi(\boldsymbol{\omega})$ fails to accurately predict the maximum expected return. This ultimately causes the guidance target $\mathbf{1}^n$ used in the deployment phase to mismatch the target corresponding to the high trajectory return in training, resulting in a suboptimal policy. The NPN method can achieve accurate normalization for multi-objective targets, significantly distinguishing the quality of trajectories in the dataset, while ensuring that the sampling targets during the training phase and the deployment phase are consistent.

## D. Proofs for Sliding Guidance

In this section, we provide a derivation process of the sliding guidance mechanism, which enables controlled continuous target distribution adjustment. Given a trained diffusion model $p_\theta$ and condition $c$, and define the adapter as $p_{\theta^*}$, our goal is to obtain $\theta^*$ that increase the likelihood of attribute $c_+$ while decrease the likelihood of "opposite" attribute $c_-$ when conditioned on $c$:

$$p_\theta\left(\boldsymbol{x}^0(\tau) \mid c\right) \leftarrow p_\theta\left(\boldsymbol{x}^0(\tau) \mid c\right) \left(\frac{p_{\theta^*}\left(c_+ \mid \boldsymbol{x}^0(\tau)\right)}{p_{\theta^*}\left(c_- \mid \boldsymbol{x}^0(\tau)\right)}\right)^\eta \tag{1}$$

where $p_\theta\left(\boldsymbol{x}^0(\tau) \mid c\right)$ represents the distribution generated by the original diffusion model $\theta$ when conditioned on $c$. According to the Bayesian formula $p\left(c_+ \mid \boldsymbol{x}^0(\tau)\right) = \frac{p\left(\boldsymbol{x}^0(\tau)\mid c_+\right)p(c_+)}{p(\boldsymbol{x}^0(\tau))}$, the gradient of the log probability $\nabla \log \left[p_\theta\left(\boldsymbol{x}^0(\tau) \mid c\right) \left(\frac{p_{\theta^*}\left(c_+\mid\boldsymbol{x}^0(\tau)\right)}{p_{\theta^*}\left(c_-\mid\boldsymbol{x}^0(\tau)\right)}\right)^\eta\right]$ can be expanded to:

$$\nabla \log \left[p_\theta\left(\boldsymbol{x}^0(\tau) \mid c\right) \left(\frac{p_{\theta^*}\left(\boldsymbol{x}^0(\tau) \mid c_+\right) p\left(c_+\right)}{p_{\theta^*}\left(\boldsymbol{x}^0(\tau) \mid c_-\right) p\left(c_-\right)}\right)^\eta\right] \tag{2}$$

which is proportional to:

$$\nabla \log p_\theta\left(\boldsymbol{x}^0(\tau) \mid c\right) + \eta\left(\nabla \log p_{\theta^*}\left(\boldsymbol{x}^0(\tau) \mid c_+\right) - \nabla \log p_{\theta^*}\left(\boldsymbol{x}^0(\tau) \mid c_-\right)\right) \tag{3}$$

Based on the reparametrization trick of (Ho et al., 2020), we can introduce a time-varying noising process and express each score (gradient of log probability) as a denoising prediction $\epsilon\left(\boldsymbol{x}^t, c, t\right)$, thus Equation (3) is equivalent to modifying the noise prediction model:

$$\epsilon_\theta\left(\boldsymbol{x}^t, c, t\right) \leftarrow \epsilon_\theta\left(\boldsymbol{x}^t, c, t\right) + \eta\left(\epsilon_{\theta^*}\left(\boldsymbol{x}^t, c_+, t\right) - \epsilon_{\theta^*}\left(\boldsymbol{x}^t, c_-, t\right)\right) \tag{4}$$

# E. Pseudo Code

In this section, we explain the training and inference process of MODULI in pseudo code. MODULI first trains a standard diffusion model for conditional generation; then, we freeze the parameters of this model and use an additional network, the slider, to learn the latent direction of policy shift. Finally, we jointly use the original diffusion and the slider to achieve controllable preference generalization. The pseudo code are given below.

**Training Process of MODULI**    The training process of the MODULI consists of two parts: the training of the diffusion model and the training of the slider adapter, which are described in Algorithm 1 and Algorithm 2, respectively. Given a normalized dataset $D_g$ containing trajectories of states $s$, normalized RTG $g$ and preferences $\omega$, MODULI first trains a diffusion planner $\theta$:

---

**Algorithm 1** Diffusion Training

---

**Require:** normalized dataset $\mathcal{D}_{\boldsymbol{g}}$, diffusion model $\theta$
  **while** not done **do**
    $(\boldsymbol{x}^0, \boldsymbol{\omega}, \boldsymbol{g}) \sim \mathcal{D}_{\boldsymbol{g}}$
    $t \sim \text{Uniform}(\{1, \cdots, T\})$
    $\boldsymbol{\epsilon} \sim \mathcal{N}(\mathbf{0}, \boldsymbol{I})$
    $\mathcal{L}(\theta) = \mathbb{E}_{(\boldsymbol{x}^0, \boldsymbol{\omega}, \boldsymbol{g}) \sim \mathcal{D}, t \sim U(0,T), \boldsymbol{\epsilon} \sim \mathcal{N}(\mathbf{0}, \boldsymbol{I})} \left[ \|\boldsymbol{\epsilon}_\theta(\boldsymbol{x}^t, t, \boldsymbol{\omega}, \boldsymbol{g}) - \boldsymbol{\epsilon}\|_2^2 \right]$
    Update $\theta$ to minimize $\mathcal{L}(\theta)$
  **end while**

---

After training the diffusion model $\theta$, we freeze its parameters to prevent any changes when training the slider. At each step, we sample a random preference shift from a uniform distribution $\Delta\boldsymbol{\omega} \sim \text{Uniform}(\{-\Delta\boldsymbol{\omega}^{\text{max}}, \Delta\boldsymbol{\omega}^{\text{max}}\})$, where $\Delta\boldsymbol{\omega}^{\text{max}}$ is a preset maximum preference shift to avoid sampling ood preference. In implementation, $\Delta\boldsymbol{\omega}^{\text{max}}$ is set to $1e-3$. We then use the diffusion model to sample a pair of predicted noises $(\boldsymbol{\epsilon}_\theta(\boldsymbol{x}^t, t, \boldsymbol{\omega} + \Delta\boldsymbol{\omega}, \boldsymbol{g}), \boldsymbol{\epsilon}_\theta(\boldsymbol{x}^t, t, \boldsymbol{\omega} - \Delta\boldsymbol{\omega}, \boldsymbol{g}))$ whose preferences are positively and negatively shifted, and divide them by the shift value of the preference $2\Delta\boldsymbol{\omega}$ to represent the "noise shift caused by unit preference shift". The slider model is used to explicitly capture this shift. The specific training process and gradient target are detailed below:

---

**Algorithm 2** Slider Training

---

**Require:** normalized dataset $\mathcal{D}_{\boldsymbol{g}}$, pretrained diffusion model $\theta$, slider adapter $\theta^*$, preference shift $\Delta\boldsymbol{\omega}^{\text{max}}$
  **while** not done **do**
    $(\boldsymbol{x}^0, \boldsymbol{\omega}, \boldsymbol{g}) \sim \mathcal{D}$
    $\Delta\boldsymbol{\omega} \sim \text{Uniform}(\{-\Delta\boldsymbol{\omega}^{\text{max}}, \Delta\boldsymbol{\omega}^{\text{max}}\})$
    $t \sim \text{Uniform}(\{1, \cdots, T\})$
    $\boldsymbol{\epsilon} \sim \mathcal{N}(\mathbf{0}, \boldsymbol{I})$
    $\mathcal{L}(\theta^*) = \mathbb{E}_{(\boldsymbol{\omega}, \boldsymbol{g}, \Delta\boldsymbol{\omega}) \sim D} \left[ \left\| \boldsymbol{\epsilon}_{\theta^*}(\boldsymbol{x}^t, t, \boldsymbol{\omega}, \boldsymbol{g}) - \frac{[\boldsymbol{\epsilon}_\theta(\boldsymbol{x}^t, t, \boldsymbol{\omega}+\Delta\boldsymbol{\omega}, \boldsymbol{g}) - \boldsymbol{\epsilon}_\theta(\boldsymbol{x}^t, t, \boldsymbol{\omega}-\Delta\boldsymbol{\omega}, \boldsymbol{g})]}{2\Delta\boldsymbol{\omega}} \right\|_2^2 \right]$
    Update $\theta^*$ to minimize $\mathcal{L}(\theta^*)$
  **end while**

---

**Inference Process of MODULI**    During deployment, MODULI utilizes both the diffusion model $\theta$ and the slider $\theta^*$ to guide the denoising process. Given a desired preference $\boldsymbol{\omega}$, we first query the closest preference $\boldsymbol{\omega}_0$ in the dataset. In the $S$-step denoising process, we expect the distribution of the generated trajectory $\boldsymbol{x}$ to gradually shift from the in-distribution preference $\boldsymbol{\omega}_0$ to the desired preference $\boldsymbol{\omega}$ through the guidance of the slider. Formally, we expect that the trajectory $\boldsymbol{x}^{\kappa_i}$ at the $i$-th denoising step falls within the distribution $p(\boldsymbol{x}^{\kappa_i} \mid \boldsymbol{\omega}_i, \kappa_i)$. After one step of diffusion denoising, it falls within the distribution $p(\boldsymbol{x}^{\kappa_{i-1}} \mid \boldsymbol{\omega}_i, \kappa_{i-1})$. Subsequently, after the slider shift, it falls within the distribution $p(\boldsymbol{x}^{\kappa_{i-1}} \mid \boldsymbol{\omega}_{i-1}, \kappa_{i-1})$. The pseudocode for the implementation is provided in Algorithm 3:

---

**Algorithm 3** MODULI planning

---

**Require:** dataset $\mathcal{D}$, diffusion model $\theta$, slider $\theta^*$, target preference $\boldsymbol{\omega}$, inverse dynamics $h_\phi$, $S$-length sampling sequence $\kappa_S$, classifier-free guidance scale $w$

$\quad \boldsymbol{\omega}_0 \leftarrow \underset{\boldsymbol{\omega}_0}{\arg\min} \|\boldsymbol{\omega} - \boldsymbol{\omega}_0\|_2^2, \; \boldsymbol{\omega}_0 \in \mathcal{D}$

$\quad \Delta\boldsymbol{\omega} = \boldsymbol{\omega} - \boldsymbol{\omega_0}$

$\quad$ **while** not done **do**

$\quad\quad$ Observe state $\boldsymbol{s}_t$; Sample noise from prior distribution $\boldsymbol{x}^{\kappa_S} \sim \mathcal{N}(\boldsymbol{0}, \boldsymbol{I})$

$\quad\quad$ **for** $i = S, \cdots, 1$ **do**

$\quad\quad\quad \boldsymbol{\omega}_i \leftarrow \boldsymbol{\omega}_0 + \frac{S-i}{S}\Delta\boldsymbol{\omega}$

$\quad\quad\quad \tilde{\epsilon}_\theta \leftarrow (1+w)\epsilon_\theta(\boldsymbol{x}^{\kappa_i}, \kappa_i, \boldsymbol{\omega}_i, \boldsymbol{1}^n) - w\epsilon_\theta(\boldsymbol{x}^{\kappa_i}, \kappa_i)$

$\quad\quad\quad \Delta\tilde{\epsilon}_{\theta^*} = \frac{\Delta\boldsymbol{\omega}}{S}\epsilon_{\theta^*}(\boldsymbol{x}^{\kappa_i}, \kappa_i, \boldsymbol{\omega}_i, \boldsymbol{1}^n)$

$\quad\quad\quad \boldsymbol{x}^{\kappa_{i-1}} \leftarrow \text{Denoise}(\boldsymbol{x}^{\kappa_i}, \tilde{\epsilon}_\theta + \Delta\tilde{\epsilon}_{\theta^*})$

$\quad\quad$ **end for**

$\quad\quad$ Extract $\boldsymbol{s}_t, \boldsymbol{s}_{t+1}$ from $\boldsymbol{x}^{\kappa_0}$ and derive $\boldsymbol{a}_t = h_\phi(s_t, s_{t+1})$

$\quad\quad$ Execute $\boldsymbol{a}_t$

$\quad$ **end while**

---

## F. Details of Loss-weight Trick

The diffusion planner generates a state sequence $x^0(\tau) = [s_0, \cdots, s_{H-1}]$. Since closed-loop control is used, only $s_0$ and $s_1$ are most relevant to the current decision. When calculating the loss, the weight of $s_1$ is increased from 1 to 10, which is considered more important. For clarity, we provide pseudocode below for the loss calculation of the diffusion model incorporating the loss-weight trick and explain it in detail.

```python
# Create a loss_weight and assign a higher
loss weight to the next observation (o_1).

next_obs_loss_weight = 10
loss_weight = torch.ones((batch_size, horizon, obs_dim))
loss_weight[:, 1, :] = next_obs_loss_weight

def diffusion_loss_pseudocode(x0, condition):
"""
Pseudocode function to demonstrate the loss function in diffusion models

Variable descriptions:
x0: Original noise-free data - [batch_size, horizon, obs_dim]
condition: Conditional information - [batch_size, condition_dim]
xt: Noisy data after noise addition - same shape as x0
t: Time points in the diffusion process step - [batch_size]
eps: Noise added to x0 - same shape as x0
loss_weight: Loss weight coefficient - scalar or tensor matching x0's shape
"""

# Step 1: Add noise to the original data
xt, t, eps = add_noise(x0) # Generate noisy xt from x0

# Step 2: Process conditional information
# Encode the original condition information
condition = condition_encoder(condition)

# Step 3: Predict noise through the diffusion model
predicted_eps = diffusion_model(xt, t, condition) # Model predicts the added noise
```

```
# Step 4: Calculate mean
squared error
# Mean squared error between predicted noise and actual noise
loss = (predicted_eps - eps) ** 2

# Step 5: Apply loss weights
# We assign a higher weight to the prediction of the s_1
loss = loss * loss_weight

# Step 6: Calculate average loss
final_loss = loss.mean() # Calculate the average loss

return final_loss
```

*Table 4.* HV and SP performance on full `Complete` datasets. (B: behavioral policy). The best scores are emphasized in bold.

| | Env | Metrics | B | MODT(P) | MORvS(P) | BC(P) | CQL(P) | MODULI |
|---|---|---|---|---|---|---|---|---|
| Amateur | Ant | HV ($\times 10^6$) ↑ | 5.61 | 5.92±.04 | 6.07±.02 | 4.37±.06 | 5.62±.23 | **6.08±.03** |
| | | SP ($\times 10^4$) ↓ | - | 8.72±.77 | 5.24±.52 | 25.90±16.40 | 1.06±.28 | **0.53±.05** |
| | HalfCheetah | HV ($\times 10^6$) ↑ | 5.68 | 5.69±.01 | **5.77±.00** | 5.46±.02 | 5.54±.02 | 5.76±.00 |
| | | SP ($\times 10^4$) ↓ | - | 1.16±.42 | 0.57±.09 | 2.22±.91 | 0.45±.27 | **0.07±.02** |
| | Hopper | HV ($\times 10^7$) ↑ | 1.97 | 1.81±.05 | 1.76±.03 | 1.35±.03 | 1.64±.01 | **2.01±.01** |
| | | SP ($\times 10^5$) ↓ | - | 1.61±.29 | 3.50±1.54 | 2.42±1.08 | 3.30±5.25 | **0.10±.01** |
| | Swimmer | HV ($\times 10^4$) ↑ | 2.11 | 1.67±.22 | 2.79±.03 | 2.82±.04 | 1.69±.93 | **3.20±.00** |
| | | SP ($\times 10^0$) ↓ | - | 2.87±1.32 | **1.03±.20** | 5.05±1.82 | 8.87±6.24 | 9.50±.59 |
| | Walker2d | HV ($\times 10^6$) ↑ | 4.99 | 3.10±.34 | 4.98±.01 | 3.42±.42 | 1.78±.33 | **5.06±.00** |
| | | SP ($\times 10^4$) ↓ | - | 164.20±13.50 | 1.94±.06 | 53.10±34.60 | 7.33±5.89 | **0.25±.03** |
| | Hopper-3obj | HV ($\times 10^{10}$) ↑ | 3.09 | 1.04±.16 | 2.77±.24 | 2.42±.18 | 0.59±.42 | **3.33±.06** |
| | | SP ($\times 10^5$) ↓ | - | 10.23±2.78 | 1.03±.11 | 0.87±.29 | 2.00±1.72 | **0.10±.00** |
| Expert | Ant | HV ($\times 10^6$) ↑ | 6.32 | 6.21±.01 | 6.36±.02 | 4.88±.17 | 5.76±.10 | **6.39±.02** |
| | | SP ($\times 10^4$) ↓ | - | 8.26±2.22 | 0.87±.19 | 46.20±16.40 | 0.58±.10 | **0.79±.12** |
| | HalfCheetah | HV ($\times 10^6$) ↑ | 5.79 | 5.73±.00 | 5.78±.00 | 5.54±.05 | 5.63±.04 | **5.79±.00** |
| | | SP ($\times 10^4$) ↓ | - | 1.24±.23 | 0.67±.05 | 1.78±.39 | 0.10±.00 | **0.07±.00** |
| | Hopper | HV ($\times 10^7$) ↑ | 2.09 | 2.00±.02 | 2.02±.02 | 1.23±.10 | 0.33±.39 | **2.09±.01** |
| | | SP ($\times 10^5$) ↓ | - | 16.30±10.60 | 3.03±.36 | 52.50±4.88 | 2.84±2.46 | **0.09±.01** |
| | Swimmer | HV ($\times 10^4$) ↑ | 3.25 | 3.15±.02 | **3.24±.00** | 3.21±.00 | 3.22±.08 | **3.24±.00** |
| | | SP ($\times 10^0$) ↓ | - | 15.00±7.49 | **4.39±.07** | 4.50±.39 | 13.60±5.31 | 4.43±.38 |
| | Walker2d | HV ($\times 10^6$) ↑ | 5.21 | 4.89±.05 | 5.14±.01 | 3.74±.11 | 3.21±.32 | **5.20±.00** |
| | | SP ($\times 10^4$) ↓ | - | 0.99±.44 | 3.22±.73 | 75.60±52.30 | 6.23±10.70 | **0.11±.01** |
| | Hopper-3obj | HV ($\times 10^{10}$) ↑ | 3.73 | 3.38±.05 | 3.42±.10 | 2.29±.07 | 0.78±.24 | **3.57±.02** |
| | | SP ($\times 10^5$) ↓ | - | 1.40±.44 | 2.72±1.93 | 0.72±.09 | 2.60±3.14 | **0.07±.00** |

## G. Full Results

See Table 4, Table 5 and Table 6 for full results of HV, RD, SP on `Complete` / `Shattered` / `Narrow` datasets.

## H. Additional Experiment Results

### H.1. Performance with Different Guidance Mechanism

As shown in Table 7, we also compare the slider with different guided sampling methods on the expert datasets, namely CG and CG+CFG. Classifier Guidance (CG) requires training an additional classifier $\log p_\phi(\boldsymbol{\omega}, \boldsymbol{g}|\boldsymbol{x}^t, t)$ to predict the log probability of exhibiting property. During inference, the classifier's gradient is used to guide the solver. CG+CFG combines the CG and CFG methods, using preference $\boldsymbol{\omega}$ for guidance in the CFG part and RTG classifier $\log p_\phi(\boldsymbol{g}|\boldsymbol{x}^t, t)$ for guidance in the CG part. We found that the CFG+CG method performs better than the CG method in most environments, indicating that CG is more suitable for guiding trajectories with high returns. However, MODULI w/ slider still demonstrates the best performance in most environments, especially in the HV and RD metrics.

### H.2. Approximate Pareto Front on all Environments

We present the approximate Pareto front visualization results of MODULI for all environments in Figure 7. MODULI is a good Pareto front approximator, capable of extending the front on the amateur dataset and performing interpolation or extrapolation generalization on the `Shattered` or `Narrow` datasets.

*Table 5.* HV, RD and SP performance on full `Expert-Shattered` and `Expert-Narrow` datasets. The best scores are emphasized in bold.

| | Env | Metrics | MODT(P) | MORvS(P) | BC(P) | MODULI |
|---|---|---|---|---|---|---|
| *Expert-Shattered* | Ant | HV ($\times 10^6$) $\uparrow$ | 5.88±.02 | 6.38±.01 | 4.76±.05 | **6.39±.01** |
| | | RD ($\times 10^3$) $\downarrow$ | 0.71±.03 | 0.16±.00 | 0.61±.01 | **0.14±.01** |
| | | SP ($\times 10^4$) $\downarrow$ | 2.22±.50 | 0.75±.13 | 5.16±1.18 | **0.53±.13** |
| | HalfCheetah | HV ($\times 10^6$) $\uparrow$ | 5.69±.00 | 5.73±.00 | 5.66±.00 | **5.78±.00** |
| | | RD ($\times 10^3$) $\downarrow$ | 0.18±.00 | 0.10±.00 | 0.28±.00 | **0.07±.00** |
| | | SP ($\times 10^4$) $\downarrow$ | 0.18±.01 | 0.15±.02 | 0.34±.03 | **0.11±.02** |
| | Hopper | HV ($\times 10^7$) $\uparrow$ | 1.95±.03 | 2.06±.01 | 1.49±.01 | **2.07±.00** |
| | | RD ($\times 10^3$) $\downarrow$ | 1.42±.00 | 1.10±.04 | 3.92±.11 | **0.22±.06** |
| | | SP ($\times 10^5$) $\downarrow$ | 0.85±.27 | 0.20±.06 | 2.58±1.01 | **0.11±.01** |
| | Swimmer | HV ($\times 10^4$) $\uparrow$ | 3.20±.00 | **3.24±.00** | 3.19±.00 | **3.24±.00** |
| | | RD ($\times 10^2$) $\downarrow$ | 0.38±.00 | 0.16±.00 | 0.07±.00 | **0.06±.00** |
| | | SP ($\times 10^0$) $\downarrow$ | 15.50±.33 | 7.36±.81 | 9.05±.30 | **5.79±.43** |
| | Walker2d | HV ($\times 10^6$) $\uparrow$ | 5.01±.01 | 5.14±.01 | 3.47±.12 | **5.20±.00** |
| | | RD ($\times 10^3$) $\downarrow$ | 0.86±.01 | 0.65±.02 | 1.72±.01 | **0.15±.01** |
| | | SP ($\times 10^4$) $\downarrow$ | 0.49±.03 | 0.27±.04 | 16.61±13.78 | **0.12±.01** |
| | Hopper-3obj | HV ($\times 10^{10}$) $\uparrow$ | 2.83±.06 | 3.28±.07 | 1.95±.16 | **3.43±.02** |
| | | RD ($\times 10^3$) $\downarrow$ | 1.60±.01 | 1.60±.02 | 3.21±.01 | **1.28±.03** |
| | | SP ($\times 10^5$) $\downarrow$ | 0.09±.01 | **0.06±.00** | 0.17±.06 | 0.10±.01 |
| *Expert-Narrow* | Ant | HV ($\times 10^6$) $\uparrow$ | 5.05±.08 | 6.06±.00 | 4.90±.02 | **6.36±.00** |
| | | RD ($\times 10^3$) $\downarrow$ | 0.98±.01 | 0.37±.03 | 0.78±.03 | **0.25±.01** |
| | | SP ($\times 10^4$) $\downarrow$ | 0.88±.41 | 0.79±.11 | 1.56±.28 | **0.80±.25** |
| | HalfCheetah | HV ($\times 10^6$) $\uparrow$ | 5.04±.00 | 5.46±.01 | 4.88±.02 | **5.76±.00** |
| | | RD ($\times 10^3$) $\downarrow$ | 0.43±.00 | 0.29±.00 | 0.44±.00 | **0.17±.00** |
| | | SP ($\times 10^4$) $\downarrow$ | 0.04±.00 | 0.05±.00 | 0.02±.00 | **0.04±.00** |
| | Hopper | HV ($\times 10^7$) $\uparrow$ | 1.85±.02 | 1.98±.00 | 1.29±.01 | **2.04±.01** |
| | | RD ($\times 10^3$) $\downarrow$ | 4.24±.02 | **1.37±.05** | 3.83±.05 | 2.42±.05 |
| | | SP ($\times 10^5$) $\downarrow$ | 0.34±.07 | **0.16±.02** | 0.59±.24 | 0.25±.05 |
| | Swimmer | HV ($\times 10^4$) $\uparrow$ | 3.03±.00 | 3.10±.00 | 3.06±.00 | **3.21±.00** |
| | | RD ($\times 10^2$) $\downarrow$ | 0.38±.00 | 0.39±.01 | 0.19±.00 | **0.10±.00** |
| | | SP ($\times 10^0$) $\downarrow$ | 3.65±.22 | 4.38±.80 | 5.53±.20 | **3.28±.15** |
| | Walker2d | HV ($\times 10^6$) $\uparrow$ | 4.75±.01 | 4.85±.01 | 1.88±.95 | **5.10±.01** |
| | | RD ($\times 10^3$) $\downarrow$ | 1.07±.01 | 1.45±.02 | 1.88±.00 | **0.28±.02** |
| | | SP ($\times 10^4$) $\downarrow$ | 0.34±.09 | 0.19±.02 | 10.76±11.58 | **0.13±.01** |
| | Hopper-3obj | HV ($\times 10^{10}$) $\uparrow$ | 3.18±.05 | 3.32±.00 | 2.32±.10 | **3.37±.05** |
| | | RD ($\times 10^3$) $\downarrow$ | 2.48±.00 | **2.24±.02** | 2.49±.02 | 2.38±.01 |
| | | SP ($\times 10^5$) $\downarrow$ | 0.11±.01 | 0.22±.02 | 0.14±.02 | **0.12±.01** |

*Table 6.* HV, RD and SP performance on full `Amateur-Shattered` and `Amateur-Narrow` datasets. The best scores are emphasized in bold.

| | Env | Metrics | MODT(P) | MORvS(P) | BC(P) | MODULI |
|---|---|---|---|---|---|---|
| *Amateur-Shattered* | Ant | HV ($\times10^6$) ↑ | 5.74±.04 | **6.09±.01** | 4.20±.05 | 6.04±.01 |
| | | RD ($\times10^3$) ↓ | 0.36±.01 | 0.13±.03 | 1.01±.07 | **0.09±.01** |
| | | SP ($\times10^4$) ↓ | 2.47±.61 | 0.79±.14 | 2.29±1.38 | **0.63±.23** |
| | HalfCheetah | HV ($\times10^6$) ↑ | 5.61±.01 | 5.74±.00 | 5.54±.00 | **5.75±.00** |
| | | RD ($\times10^3$) ↓ | 0.62±.00 | 0.11±.00 | 0.11±.00 | **0.07±.00** |
| | | SP ($\times10^4$) ↓ | 0.11±.01 | **0.09±.01** | **0.09±.01** | 0.13±.01 |
| | Hopper | HV ($\times10^7$) ↑ | 1.46±.00 | 1.69±.01 | 1.63±.00 | **2.01±.01** |
| | | RD ($\times10^3$) ↓ | 2.51±.01 | 1.76±.02 | 0.99±.06 | **0.49±.15** |
| | | SP ($\times10^5$) ↓ | 0.54±.58 | 0.18±.01 | 0.11±.05 | **0.15±.04** |
| | Swimmer | HV ($\times10^4$) ↑ | 0.75±.00 | 2.81±.00 | 2.81±.00 | **3.17±.00** |
| | | RD ($\times10^2$) ↓ | 0.94±.00 | 0.40±.00 | 0.35±.00 | **0.29±.00** |
| | | SP ($\times10^0$) ↓ | 6.75±.23 | **1.26±.06** | 1.89±.05 | 8.85±.77 |
| | Walker2d | HV ($\times10^6$) ↑ | 3.86±.07 | 4.99±.00 | 3.42±.02 | **5.03±.00** |
| | | RD ($\times10^3$) ↓ | 1.44±.02 | 0.37±.02 | 1.18±.01 | **0.35±.02** |
| | | SP ($\times10^4$) ↓ | 11.13±3.58 | **0.20±.03** | 4.56±.08 | 0.29±.01 |
| | Hopper-3obj | HV ($\times10^{10}$) ↑ | 1.48±.02 | 2.84±.04 | 1.34±.02 | **3.02±.03** |
| | | RD ($\times10^3$) ↓ | 2.86±.01 | 1.29±.01 | 2.87±.00 | **1.01±.04** |
| | | SP ($\times10^5$) ↓ | 0.56±.08 | 0.13±.01 | 0.59±.08 | **0.10±.01** |
| *Amateur-Narrow* | Ant | HV ($\times10^6$) ↑ | 5.17±.03 | 5.80±.03 | 4.19±.10 | **5.88±.03** |
| | | RD ($\times10^3$) ↓ | 0.77±.01 | **0.27±.02** | 1.93±.04 | 0.53±.00 |
| | | SP ($\times10^4$) ↓ | **0.43±.10** | 0.51±.08 | 4.42±1.44 | 0.62±.11 |
| | HalfCheetah | HV ($\times10^6$) ↑ | 4.29±.00 | 5.55±.00 | 5.08±.01 | **5.71±.00** |
| | | RD ($\times10^3$) ↓ | 0.98±.01 | 0.36±.00 | 0.36±.00 | **0.35±.00** |
| | | SP ($\times10^4$) ↓ | 1.02±.44 | **0.02±.00** | **0.02±.03** | 0.06±.02 |
| | Hopper | HV ($\times10^7$) ↑ | 1.73±.01 | 1.69±.01 | 0.94±.34 | **1.99±.00** |
| | | RD ($\times10^3$) ↓ | 1.38±.04 | 2.88±.08 | 5.22±.01 | **1.33±.11** |
| | | SP ($\times10^5$) ↓ | 0.27±.07 | 0.25±.03 | 5.81±8.21 | **0.22±.06** |
| | Swimmer | HV ($\times10^4$) ↑ | 0.66±.03 | 2.95±.00 | 2.80±.00 | **3.15±.00** |
| | | RD ($\times10^2$) ↓ | 1.14±.00 | 0.48±.00 | 0.49±.00 | **0.45±.00** |
| | | SP ($\times10^0$) ↓ | 14.25±3.73 | **1.91±.01** | 2.95±.08 | 5.79±.43 |
| | Walker2d | HV ($\times10^6$) ↑ | 3.24±..16 | 4.83±.00 | 2.83±.01 | **4.91±.01** |
| | | RD ($\times10^3$) ↓ | 1.91±.00 | 1.77±.00 | 1.87±.00 | **0.97±.02** |
| | | SP ($\times10^4$) ↓ | 5.64±3.72 | 0.22±.01 | **0.01±.00** | 0.24±.01 |
| | Hopper-3obj | HV ($\times10^{10}$) ↑ | 1.85±.04 | 2.27±.05 | 2.26±.02 | **2.90±.05** |
| | | RD ($\times10^3$) ↓ | 2.44±.02 | **1.60±.02** | 1.86±.03 | 1.84±.04 |
| | | SP ($\times10^5$) ↓ | 3.12±1.21 | **0.07±.01** | 0.09±.00 | 0.12±.02 |

*Table 7.* HV, RD and SP performance with different guided sampling methods on the expert datasets.

| | Env | Metrics | MODULI | CG | CFG+CG |
|---|---|---|---|---|---|
| *Expert-Narrow* | Ant | HV ($\times 10^6$) ↑ | **6.36±.00** | 6.23±.03 | 6.12±.03 |
| | | RD ($\times 10^3$) ↓ | **0.25±.01** | 0.44±.03 | 0.85±.03 |
| | | SP ($\times 10^4$) ↓ | **0.80±.25** | 0.52±.16 | 0.81±.12 |
| | HalfCheetah | HV ($\times 10^6$) ↑ | **5.76±.00** | 5.69±.01 | **5.76±.00** |
| | | RD ($\times 10^3$) ↓ | 0.17±.00 | **0.14±.03** | 0.52±.03 |
| | | SP ($\times 10^4$) ↓ | **0.04±.00** | 0.47±.03 | **0.04±.00** |
| | Hopper | HV ($\times 10^7$) ↑ | **2.04±.01** | 1.95±.01 | 2.03±.01 |
| | | RD ($\times 10^3$) ↓ | 2.42±.05 | **1.11±.14** | 2.97±.05 |
| | | SP ($\times 10^5$) ↓ | 0.25±.05 | **0.09±.01** | **0.09±.02** |
| | Swimmer | HV ($\times 10^4$) ↑ | **3.21±.00** | 3.19±.01 | 3.16±.00 |
| | | RD ($\times 10^2$) ↓ | **0.10±.00** | 0.16±.01 | 1.12±.00 |
| | | SP ($\times 10^0$) ↓ | **3.28±.15** | 23.28±3.56 | 5.26±.62 |
| | Walker2d | HV ($\times 10^6$) ↑ | **5.10±.01** | 5.01±.06 | 5.08±.00 |
| | | RD ($\times 10^3$) ↓ | **0.28±.02** | 0.40±.04 | 0.63±.00 |
| | | SP ($\times 10^4$) ↓ | **0.13±.01** | 1.27±.32 | 0.19±.00 |
| | Hopper-3obj | HV ($\times 10^{10}$) ↑ | **3.37±.05** | 2.30±.08 | 3.34±.03 |
| | | RD ($\times 10^3$) ↓ | **2.38±.01** | 2.58±.04 | 2.45±.03 |
| | | SP ($\times 10^5$) ↓ | 0.12±.01 | **0.11±.01** | **0.11±.01** |
| *Expert-Shattered* | Ant | HV ($\times 10^6$) ↑ | **6.39±.01** | 6.23±.03 | 6.36±.03 |
| | | RD ($\times 10^3$) ↓ | **0.14±.01** | 0.46±.02 | **0.14±.00** |
| | | SP ($\times 10^4$) ↓ | **0.53±.13** | 0.54±.13 | 0.54±.16 |
| | HalfCheetah | HV ($\times 10^6$) ↑ | **5.78±.00** | 5.69±.01 | **5.78±.00** |
| | | RD ($\times 10^3$) ↓ | **0.07±.00** | 0.37±.00 | **0.07±.03** |
| | | SP ($\times 10^4$) ↓ | **0.11±.02** | 0.47±.03 | 0.12±.01 |
| | Hopper | HV ($\times 10^7$) ↑ | **2.07±.01** | 1.95±.01 | **2.07±.01** |
| | | RD ($\times 10^3$) ↓ | **0.22±.06** | 0.81±.19 | 0.23±.03 |
| | | SP ($\times 10^5$) ↓ | **0.11±.01** | 0.13±.12 | 0.12±.01 |
| | Swimmer | HV ($\times 10^4$) ↑ | **3.24±.00** | 3.19±.01 | **3.24±.00** |
| | | RD ($\times 10^2$) ↓ | 0.06±.00 | 0.29±.02 | **0.04±.00** |
| | | SP ($\times 10^0$) ↓ | **5.79±.43** | 23.28±3.56 | 6.78±1.26 |
| | Walker2d | HV ($\times 10^6$) ↑ | **5.20±.00** | 5.19±.00 | 5.01±.06 |
| | | RD ($\times 10^3$) ↓ | **0.15±.01** | 0.17±.01 | 0.73±.04 |
| | | SP ($\times 10^4$) ↓ | **0.12±.01** | **0.12±.01** | 1.27±.32 |
| | Hopper-3obj | HV ($\times 10^{10}$) ↑ | **3.43±.02** | 2.30±.08 | 3.39±.02 |
| | | RD ($\times 10^3$) ↓ | **1.28±.03** | 2.45±.10 | 1.66±.01 |
| | | SP ($\times 10^5$) ↓ | **0.10±.01** | 0.12±.01 | **0.10±.01** |

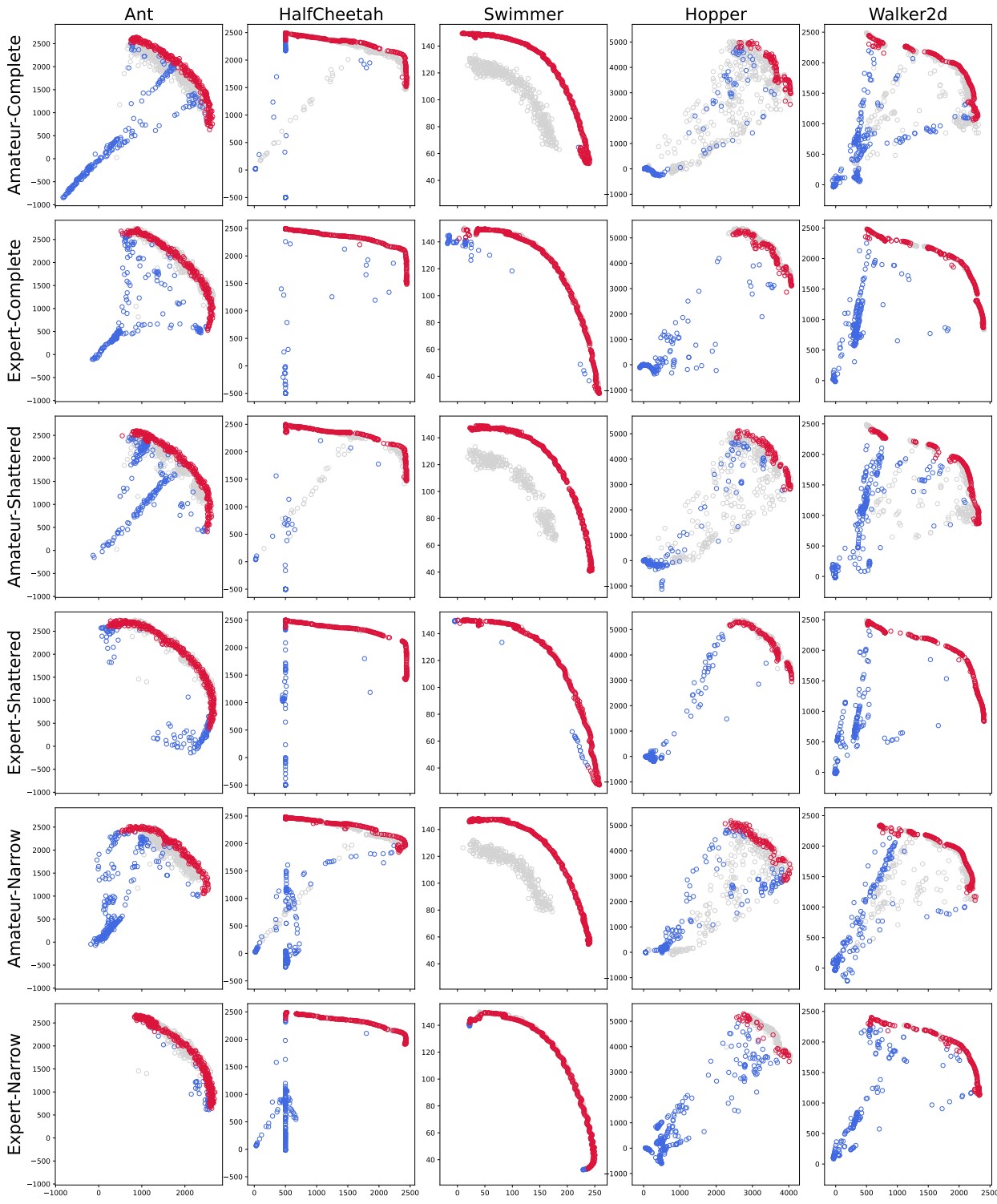

*Figure 7.* Approximated Pareto fronts on `Complete` / `Shattered` / `Narrow` dataset by MODULI. Undominated / Dominated solutions are colored in red / blue, and dataset trajectories are colored in grey.

