# OpenReview forum: "MODULI: Unlocking Preference Generalization via Diffusion Models for Offline Multi-Objective Reinforcement Learning"
_ICML.cc/2025/Conference — ICML 2025 poster_

### Official Review · Reviewer_Dt55 · 2025-02-15

**Overall Recommendation:** 4

**Summary:**

This paper introduces MODULI (Multi-Objective Diffusion planner with sliding guidance), a novel algorithm for offline multi-objective reinforcement learning (MORL). MODULI employs a conditional diffusion model for representing the agent’s policy. Besides MODULI, the paper also introduces two techniques for normalizing multi-objective returns that are used to condition the policy, and a “slider adapter” neural network used to make the diffusion model generalize to unseen OOD preferences.

---

### Post Rebuttal

I thank the authors once again for providing detailed answers to my questions.

I increased my score since the authors addressed my raised concerns.

**Claims And Evidence:**

The claims are evaluated empirically by evaluating the method in offline datasets with missing examples for some regions of preferences, or with narrow distribution of solutions in the Pareto frontier. The empirical results seem to be in accordance with the claims. That is, 1) the proposed method achieves generally better metrics than the baselines, 2) the addition of the slider adapter network seems to indeed improve the generalization of the method, and 3) the normalization techniques introduced improve the method when compared to the global normalization baseline.

However, in Figure 3 (bottom), the authors claim that “MODULI can significantly extend the Pareto front to both sides”. However, this is not true if we inspect the plot. MODULI can reproduce the solutions already in the dataset on the sides, but it does not generalize to more extreme points.

Although the results seem to validate the claim, the performance improvements compared to the MORvS(P) baseline are not that significant.

**Essential References Not Discussed:**

The authors discussed all previous relevant papers in the field of offline MORL.

**Experimental Designs Or Analyses:**

I checked the details of the experimental designs. One issue with the paper is that, for each experiment, the authors used only 3 random seeds to report the mean and standard deviation. The authors should justify why only 3 random seeds are enough.

**Methods And Evaluation Criteria:**

The method is compared with other recent offline algorithms in the D4MORL benchmark, which is a recent Mujoco benchmark for offline MORL. They constitute appropriate baselines and benchmarks for evaluating the method.

**Other Comments Or Suggestions:**

* A few references are citing the ArXiv version of papers that have been published in peer-reviewed venues, e.g., “Alegre et al. Sample-efficient multi-objective learning via generalized policy improvement prioritization. arXiv preprint arXiv:2301.07784, 2023” has been published at AAMAS 2023.

* ”As a result, direct guidance using the target conditions of $y = [\omega_{target}, 1^n]$ may be unachievable for some preferences, leading to a performance drop.”
Since this is a multi-objective setting, the target $1^n$ will always be unachievable. The vector $1^n$ is only achievable if all objectives are aligned/not conflicting,

* “This enables controlled continuous concept during generation”
It is not clear what a “controlled continuous concept” is.

* Typo: ”and P demotes Pareto front” -> denotes

**Other Strengths And Weaknesses:**

Below, I point out a few other weaknesses in the paper:

* The Introduction has a few technical terms that are not explained, making it confusing to understand the contributions. For instance, what is a “slider adapter”, “latent directions of preference”, or “refine guidance”? Please explain such terms in more detail.
* The paper has a few grammatical issues and sentences that are not flowing well. I suggest the authors review the text for such issues.
* Section 4.3 is currently very difficult to follow. I strongly suggest the authors provide intuition for Equation 11 and the associated variables. For instance, in the sentence “we can derive a direct fine-tuning scheme that is equivalent to modifying the noise prediction model.”, do you mean modifying the noise prediction model to achieve what goal? Also, the variable $\eta$ was not defined.
* The paper does discuss its limitations in sufficient detail. For instance, in the conclusion the authors could better state the necessary assumptions and when they are not valid.

**Questions For Authors:**

* “MODULI also employs a loss-weight trick, setting a higher weight for the next state s1 of the trajectory $x0(\tau)$ to encourage more focus on it, closely related to action execution.”
This sentence is not clear, please elaborate on how this is implemented.

* It is not clear how Equation 7 is implemented. In particular, how is each $(\omega, g)$ pair constructed? Just before the Equation, $D_P$ is defined as a set of trajectories, but then the method is sampling pairs of weights and returns. How do you ensure that each $\hat{g}$ in $D_P$ is actually the maximum return for each $\omega$?

* It is not clear how the Return Deviation (RD) metric is employed for the baseline algorithms. A different generalized return predictor is learned for each algorithm? When algorithm A has a better RD metric than an algorithm B, why does it imply better generalization?

* “However, there are limitations, such as experiments conducted in structured state space and continuous actions.”
What are structured state space and continuous actions?

**Relation To Broader Scientific Literature:**

The paper tries to improve upon previous offline MORL approaches by employing different techniques that are related to diffusion models for sequential decision-making. Prior works have employed other supervised learning techniques to tackle this problem, and the paper tries to tackle the problem with a different set of techniques.

**Theoretical Claims:**

The paper does not introduce any theorems or theoretical results.

---

> ### Author Rebuttal · Authors · 2025-04-01
>
> ## Q1: Comparison with the performance of MORvS
>
> MODULI consistently outperforms the strong MORvS baseline, leading in 20/24 metrics across 12 Complete datasets(Tab. 4) and surpassing the baseline in 59/72 metrics on Shattered & Narrow datasets(Tab. 5 & 6).
>
> ## Q2: Why use 3 seeds?
>
> We collect 501 sample points for each seed to calculate the metrics, fully covering the preference space, which results in small variances. At the same time, we aligned with the setting of D4MORL benchmark, it also uses 3 seeds. We conducted a large number of performance and ablation experiments(2 levels * 3 types * 6 tasks), demonstrating the reliability of the MODULI.
>
> ## Q3: Significantly extend the Pareto front
>
> Due to task constraints, the Pareto front in Fig.3 can only expand by a small margin. Compared to other methods, MODULI demonstrates stronger generalization and an "expansion" phenomenon. To avoid misunderstanding, we replace "significantly" → "slightly" in revision. Additionally, extensive experiments and visualization(fig.7) proves that MODULI can enhance generalization.
>
> ## Q4: Confusing expressions, incomplete issue and typos
>
> We apologise for the unclear description in the introduction. We will provide a brief introduction to the concept in the Introduction and explain them in detail in the Method. And we have carefully reviewed your suggestions part. Thank you for pointing them out! We will thoroughly review the paper, check for grammatical issues and verify the references in revision.
>
> ## Q5: Intuition for Equation 11
>
> We trained a diffusion model $p$ to generate trajectories corresponding to ID preferences. Then, we attempt to learn the pattern of change (for example, when the preference shifts from energy efficiency to high speed, the amplitude of the Swimmer's movements gradually increases). Therefore, our goal is to learn an adapter $p^*$, which applies this pattern to OOD preferences to achieve better generalization. In Equ. 11, the exponential term represents increasing the likelihood of preference $c+$ and decreasing the likelihood of preference $c−$. η Represents the guidance strength weight in classifier-free guidance for the diffusion model. Based on Equ. 11, we can use an adapter-style method to adjust the original noise prediction model ($p$). Now, the noise prediction combines the noise from the original model and the adapter.
>
> We apologize again for the confusing expressions and undefined symbols. We will provide definitions and include thorough intuitive explanations.
>
> ## Q6: Detailed limitations
>
> We assume a linear preference space, such as $w_1 + w_2 = 1$, which is the standard practice in most MORL papers. When the linear assumption does not hold, our core method remains applicable, but additional representations are required. And very sorry for the confusion. We will change the “structured state space” to “low-dimensional state space” (Distinguished from image input). “Continuous action” should be “continuous action space”, distinguished from discrete action space. We believe that MODULI can support tasks with these new modalities with minimal additional adaptation (e.g., encoder). We left it for future work.
>
> ## Q7:  Loss-weight trick
>
> Sorry for the confusion, we will include detailed implementations in the appendix. The diffusion planner generates a state sequence $x^0(\tau) = \left[ s_0, \cdots, s_{H-1} \right]$. Since closed-loop control is used, only $s_0$ and $s_1$  are most relevant to the current decision. When calculating the loss, the weight of $s_1$ is increased from 1 to 10, which is considered more important.
>
> ## Q8: Compute Equation 7
>
> In D4MORL dataset, each trajectory contains a target preference $\omega$, and $g$ is the return of the trajectory. Since $D_p$ consists of trajectories corresponding to Pareto front points, it represents the optimal level of policy in the dataset. Therefore, for each $(g, \omega)$ sampled from $D_p$,  $g$ is the maximum return corresponding to $\omega$. Then, a predictor $R(\omega)$ for the maximum possible return can be obtained with Equ.7. It can be seen as a fit to the Pareto front in the dataset.
>
> ## Q9: RD metric
>
> **Please note that we trained a predictor for each task (e.g., ant-expert) to evaluate RD, rather than one for each algorithm**. The detailed process: we fixed the Ant-**expert-complete** dataset to trained $R_\psi$. Since expert and complete trajectories were used, we can estimate the optimal $R$ for each sampled preference $w$, these ground truth information is only used for evaluation. Then, for **Ant-expert-{complete/narrow/shattered}**, for the trajectories sampled from OOD preference points, RD is defined as the difference between the ground truth return from $R_\psi$ and the actual obtained return. The results are averaged over all OOD preference points. A smaller RD indicates that the algorithm can better align with the target preference when the target preference is OOD, thus demonstrating better generalization ability.

---

> > ### Comment · Reviewer_Dt55 · 2025-04-01
> >
> > I thank the authors for their response.
> >
> > The point regarding Q7: Loss-weight trick is still unclear to me. Can the authors mathematically define such loss and its weighting terms?
> >
> > Moreover, the authors should discuss the limitations regarding the fact that their method employs closed-loop control, in contrast to standard TD-learning RL methods.

---

> > > ### Author Response · Authors · 2025-04-02
> > >
> > > Thank you for your fast responses and efforts to improve the paper! Define the current state as $s_t$, when calculating the loss function, the loss-weight trick assigns a higher weight to the state prediction at the next step $s_{t+1}$, . This is because only the predicted next state step $s_{t+1}$, and the current state step $s_t$, directly participate in the inverse dynamic model $a_t=h(s_t, s_{t+1})$. For clarity, we provide pseudocode below for the loss calculation of the diffusion model incorporating the loss-weight trick and explain it in detail. Thank you for pointing this out, we will include a detailed explanation in the appendix of the revised version.
> > >
> > > ```python
> > > ...
> > >
> > > # Create a loss_weight and assign a higher loss weight to the next observation (o_1).
> > > next_obs_loss_weight = 10
> > > loss_weight = torch.ones((batch_size, horizon, obs_dim))
> > > loss_weight[:, 1, :] = next_obs_loss_weight
> > >
> > > def diffusion_loss_pseudocode(x0, condition):
> > >     """
> > >     Pseudocode function to demonstrate the loss function in diffusion models
> > >
> > >     Variable descriptions:
> > >     x0: Original noise-free data - [batch_size, horizon, obs_dim]
> > >     condition: Conditional information - [batch_size, condition_dim]
> > >     xt: Noisy data after noise addition - same shape as x0
> > >     t: Time points in the diffusion process step - [batch_size]
> > >     eps: Noise added to x0 - same shape as x0
> > >     loss_weight: Loss weight coefficient - scalar or tensor matching x0's shape
> > >     """
> > >
> > >     # Step 1: Add noise to the original data
> > >     xt, t, eps = add_noise(x0)  # Generate noisy xt from x0
> > >
> > >     # Step 2: Process conditional information
> > >     condition = condition_encoder(condition)  # Encode the original condition information
> > >
> > >     # Step 3: Predict noise through the diffusion model
> > >     predicted_eps = diffusion_model(xt, t, condition)  # Model predicts the added noise
> > >
> > >     # Step 4: Calculate mean squared error
> > >     loss = (predicted_eps - eps) ** 2  # Mean squared error between predicted noise and actual noise
> > >
> > >     # Step 5: Apply loss weights
> > >     # We assign a higher weight to the prediction of the s_1
> > >     loss = loss * loss_weight
> > >
> > >     # Step 6: Calculate average loss
> > >     final_loss = loss.mean()  # Calculate the average loss
> > >
> > >     return final_loss
> > > ```
> > >
> > > In comparison with TD-learning, after a careful review, **we regret to point out that a clerical error occurred in our response to Q7: Loss-weight trick.** We mistakenly used the term **"closed-loop control"** to describe decision process of MODULI. Below, we describe our decision-making process in detail and compare it with the TD-Learning:
> > >
> > > **Ours**: Given the current state $s_t$, we use a diffusion model to generate the $x^0(\tau) = \left[ s_t, \cdots, s_{t+H-1} \right]$. Using $s_t$ and $s_{t+1}$, we calculate the action $a_t=h(s_t, s_{t+1})$ that needs to be executed. After executing $a_t$, the next state $s_{t+1}$ is obtained, and the above process is repeated. The diffusion model does not adjust based on environmental feedback when generating trajectories, so it operates as **an open-loop control system**.
> > >
> > > **TD-Learning Policy**: Given the current state $s_t$, the policy $\pi(a_t|s_t)$ directly outputs the action $a_t$, which is then executed to obtain $s_{t+1}$, and the process is repeated.
> > >
> > > Therefore, **the environmental information used during the decision-making phase is completely consistent** between the two approaches. However, from the perspective of sequence modeling, planning by generating trajectories and then making decisions results in a lower decision frequency compared to TD learning policy. We consider this a limitation and will incorporate a revised version—thank you for your suggestion! In the MODULI implementation, we made a simple attempt at this. We used smaller model size and fewer sampling steps, achieving a decision frequency of 10-20 Hz while maintaining high performance. Further exploration of decision frequency is left for future work.
> > >
> > > ---
> > >
> > > We hope our replies have addressed your concerns. We are always willing to answer any of your concerns about our work and we are looking forward to more inspiring discussions.

---

### Official Review · Reviewer_MyRm · 2025-03-12

**Overall Recommendation:** 4

**Summary:**

This work proposes MODULI(Multi Objective DiffUsion planner with sLIding guidance), which employs a preference-conditioned diffusion model as a planner to generate trajectories that align with various preferences and derive action for decision making. MODULI also introduces two techniques 1) new return normalization method 2) slider adapter to achieve better generation for both ID and OOD preferences. Extensive experiments on the D4MORL benchmark demonstrate superiority of MODULI.

## update after rebuttal
The authors' responses addressed most of my concerns so I've raised my score from 3 to 4.

**Claims And Evidence:**

Yes. The article clearly reflects the main arguments and the experiments can support the authors' claims.

**Essential References Not Discussed:**

No.

**Experimental Designs Or Analyses:**

Yes. The experimental designs are reasonable and can support the authors' claims.

**Methods And Evaluation Criteria:**

Yes.

**Other Comments Or Suggestions:**

No.

**Other Strengths And Weaknesses:**

#### Strengths

1. The article is well-written, with clear expression and logic, effectively reflecting the main argument.
2. The experiments are very comprehensive. The authors validate the effectiveness of their method across a series of tasks including 'complete' datasets for expressiveness evaluation and 'shattered and narrow' datasets for generalization evaluation.

#### Weaknesses

1. I think the author should clearly specify the parameter sizes of this algorithm and the other baseline algorithms for comparison, and try to keep the number of parameters consistent across different algorithms as much as possible. This is because diffusion-based methods may benefit from a larger number of parameters, and doing so will make the results more convincing.
2. I think the innovation of this work may be somewhat insufficient because the main diffusion-based planner is basically equivalent to the Decision Diffuser, and the proposed sliding guidance also draws on the work of predecessors. However, considering that these components are quite practical and have effectively improved the performance of the algorithm, I think it still meets the threshold for acceptance.

**Questions For Authors:**

No.

**Relation To Broader Scientific Literature:**

N/A

**Theoretical Claims:**

There are no theoretical claims in this paper.

---

> ### Author Rebuttal · Authors · 2025-03-31
>
> We thank the reviewer for the insightful and useful feedback, please see the following for our response.
>
> ### **Q1: Comparison of Parameters for Different Baseline Algorithms**
>
> Thank you for your suggestion! We would like to clarify that in our experiments, we standardized the parameter size of all baseline algorithms to the same magnitude as much as possible. A complete list is provided in the table below:
>
> | **Algo.** | **Model Size** |
> | --- | --- |
> | MORvS | 8.55M |
> | MOBC | 8.83M |
> | MODT | 9.75M |
> | MODULI Diffusion Model | 11.14M |
> | MODULI Inverse Dynamic Model | 1.11M |
>
> We believe your suggestion is very correct, and we will include this comparison in the revised version.
>
> ### Q2: Novelty of MODULI
>
> We would like to clarify that MODULI is not merely an application of diffusion models in offline RL. The novelty of MODULI lies in the following three aspects:
>
> - **A new research problem**: When only incomplete offline data (narrow/shattered) is available, the preference for generalization to OOD is crucial for multi-objective RL. For the first time, **We model Offline MORL from the perspective of generative models** and introduce diffusion models into offline MORL. We found that their expressive and generalization capabilities are highly effective for MORL problems. Relying solely on Decision Diffuser[1] cannot achieve the performance of MODULI, the following two techniques are important.
> - **Improved return normalization methods**: Traditional return normalization guided by the only maximum reward has critical flaws in scenarios with conflicting multi-objectives. We designed two **return normalization methods (NPN/PPN) tailored for multi-objective problems**, which are crucial for diffusion models in MORL.
> - **Better OOD Preference Generalization:** Through the study of multi-objective task datasets, we found that as preferences change, policies in multi-objective tasks will show certain changing trends, suggesting a potential direction for addressing the challenge of policy generalization in multi-objective optimization tasks. Inspired by prior work, we introduced a sliding adapter to enhance generalization capability, achieving significant results while many implementation details differed from [2], including the fine-tuning method, the calculating unit strength and the final integration of the outputs from two models. We focus on stronger generalization ability rather than precise conceptual control over new attributes.
>
> [1] Ajay A et al. Is conditional generative modeling all you need for decision-making. ICLR2023.
>
> [2] Gandikota et al. Concept Sliders: LoRA Adaptors for Precise Control in Diffusion Models. ECCV2024.

---

### Official Review · Reviewer_DFCw · 2025-03-14

**Overall Recommendation:** 3

**Summary:**

The paper studies the problem of offline multi-objective RL with preferences over the objectives. The key contribution of the paper is to introduce a new preference-conditioned diffusion model to generate trajectories aligned with specific preferences and derive actions accordingly. The generation process is enhanced with return normalization (across multiple objectives with different value scales and slide-guidance to learn preference direction to handle OOD preferences.

Experiments are conducted using the D4MORL benchmark dataset, showing the proposed method outperforms baselines such as BC, MORvS, MODT, and MOCQL.

**Claims And Evidence:**

The advantages of the proposed method are well supported by a comprehensive experimental analysis.

**Essential References Not Discussed:**

I'm not aware of any missing essential references.

**Experimental Designs Or Analyses:**

The experimental design of the paper is quite standard in MORL with commonly used evaluation criteria and benchmark datasets and state-of-the-art baselines.

**Methods And Evaluation Criteria:**

--- The D4MORL dataset used in the paper is a benchmark dataset used in multi-objective RL, which contains different types of data quality including expert and amateur datasets.
 --- The evaluation focuses on various metrics including hyper volume, sparsity, and return deviation, which are commonly used metrics in MORL.
---The idea of applying diffusion models with two enhanced techniques (return normalization and slide guidance) are well justified.

**Other Comments Or Suggestions:**

I don't have other comments or suggestions.

**Other Strengths And Weaknesses:**

Diffusion models have been used extensively in offline reinforcement learning, which is not something new. However, the strength of the paper lies on the performance enhancement obtained though return normalization and slide guidance which enable the generation process to align with preferences over multiple objectives.

**Questions For Authors:**

--- How does the level of conflict between multiple objectives impact the performance of our proposed method?
--- Most of the evaluations are for the two-objective RL tasks. Can you comment on how your method would perform when the number of objective increases?

**Relation To Broader Scientific Literature:**

Findings of the paper are related to the RL research community.

**Theoretical Claims:**

The proof for sliding guidance is provided in the appendix and looks correct.

---

> ### Author Rebuttal · Authors · 2025-03-31
>
> We thank the reviewer for the insightful and useful feedback, please see the following for our response.
>
> ### Q1: Novelty of MODULI
>
> We would like to clarify that MODULI is not merely an application of diffusion models in offline RL. The novelty of MODULI lies in the following three aspects:
>
> - **A new research problem**: When only incomplete offline data (narrow/shattered) is available, the preference for generalization to OOD is crucial for multi-objective RL. For the first time, **We model Offline MORL from the perspective of generative models** and introduce diffusion models into offline MORL. We found that their expressive and generalization capabilities are highly effective for MORL problems.
> - **Improved return normalization methods**: Traditional return normalization guided by the only maximum reward has critical flaws in scenarios with conflicting multi-objectives. We designed two **return normalization methods (NPN/PPN) tailored for multi-objective problems**, which are crucial for diffusion models in MORL.
> - **Better OOD Preference Generalization:** Through the study of multi-objective task datasets, we found that as preferences change, policies in multi-objective tasks will show certain changing trends, suggesting a potential direction for addressing the challenge of policy generalization in multi-objective optimization tasks. Inspired by prior work, we introduced a sliding adapter to enhance generalization capability, achieving significant results while many implementation details differed from [1], including the fine-tuning method, the calculating unit strength, and the final integration of the outputs from two models. We focus on stronger generalization ability rather than precise conceptual control over new attributes.
>
> ### Q2: The impact of the degree of conflict between multiple objectives on the MODULI performance
>
> We conducted a comprehensive evaluation on datasets of **varying quality (expert, amateur)** and datasets with **different levels of OOD (complete, narrow, shattered)**, all of which have varying degrees of conflicting objectives, such as speed and energy. As shown in Fig. 3 (page 7), we were surprised to find that when the dataset is incomplete, baseline algorithms like BC fail completely under OOD preferences due to conflicts, and RvS tends to fit a single solution regardless of different preferences. In contrast, MODULI adapts to varying levels of conflict and dataset quality, demonstrating excellent performance.
>
> ### Q3: Can you comment on how your method would perform when the number of objective increases?
>
> - Our experiments include the **hopper-3obj** dataset, which involves 3 conflicting objectives. It is worth noting that as the number of objectives increases, the policy for given preference becomes more complex. MODULI demonstrates outstanding performance in the three-objective task, achieving over a **20% improvement** compared to the strongest baseline MORvS **(Table 1)**.
> - From the perspective of method design, MODULI does not assume any specific number of objectives. The proposed new normalization methods and sliding adapter can handle scenarios with more objectives.
>
> [1] Gandikota et al. Concept Sliders: LoRA Adaptors for Precise Control in Diffusion Models

---

### Decision · Program_Chairs · 2025-05-01

**Decision:**

Accept (poster)

**Comment:**

This paper introduces a new algorithm called MODULI for offline multi-objective reinforcement learning. The proposed method uses a conditional diffusion model to represent the agent’s policy and leverages two new techniques—one to normalize multi-objective returns and another (a “slider adapter” neural network) to enable the diffusion model to better generalize to unseen, out-of-distribution preferences. The authors evaluated their method through extensive experiments on the D4MORL benchmark dataset and showed that MODULI outperforms several relevant baselines.

All reviewers agree that this is a well-written paper and that the authors clearly and convincingly presented, motivated, and justified their design choices and claims. Two reviewers noted that the experiments are comprehensive and demonstrate the method’s effectiveness across a wide range of qualitatively different tasks. Another reviewer commented that, although diffusion models have been extensively studied in offline RL, this work makes a substantial contribution through its novel normalization scheme and its “slider adapter” component—designed, as mentioned earlier, to support better generalization. They noted that these contributions enable significant performance improvements. Reviewers also pointed out that all theoretical results and proofs appear sound and correct. Finally, all reviewers agree that the paper’s key claims were adequately supported.

Three key concerns were raised during the discussion phase:

1. The paper’s novelty might be limited since using a diffusion-based planner could be viewed as functionally equivalent to deploying a Decision Diffuser;

2. The authors’ claim in Figure 3—that “*MODULI can significantly extend the Pareto front to both sides*”—does not appear to be accurate, and the performance improvements over the MORvS(P) baseline are not substantial;

3. The authors use only three random seeds in each experiment, which limits the statistical significance of the results, and a justification is needed for why this is sufficient to support their claims.

The authors provided a thorough rebuttal addressing all three concerns, and two reviewers increased their scores as a result.

Overall, reviewers agree that this paper contributes important insights that will benefit the ICML community and that the concerns raised during the discussion phase were not critical and can be addressed in a revised version. The reviewers encouraged the authors to improve the paper by incorporating the points raised in the reviews and discussion phase, and argued that doing so would substantially strengthen the quality and impact of the work.